# Mitochondrial translation deficiency impairs NAD+-mediated lysosomal acidification

Mikako Yagi[1,2], Takahiro Toshima[1], Rie Amamoto[1,3], Yura Do[1], Haruka Hirai[1,2], Daiki Setoyama[1], Dongchon Kang[1] & Takeshi Uchiumi[1,2,*] (iD)

## Abstract

Mitochondrial translation dysfunction is associated with neurodegenerative and cardiovascular diseases. Cells eliminate defective mitochondria by the lysosomal machinery via autophagy. The relationship between mitochondrial translation and lysosomal function is unknown. In this study, mitochondrial translation-deficient hearts from p32-knockout mice were found to exhibit enlarged lysosomes containing lipofuscin, suggesting impaired lysosome and autolysosome function. These mice also displayed autophagic abnormalities, such as p62 accumulation and LC3 localization around broken mitochondria. The expression of genes encoding for nicotinamide adenine dinucleotide (NAD+) biosynthetic enzymes—Nmnat3 and Nampt—and NAD+ levels were decreased, suggesting that NAD+ is essential for maintaining lysosomal acidification. Conversely, nicotinamide mononucleotide (NMN) administration or Nmnat3 overexpression rescued lysosomal acidification. Nmnat3 gene expression is suppressed by HIF1α, a transcription factor that is stabilized by mitochondrial translation dysfunction, suggesting that HIF1α-Nmnat3-mediated NAD+ production is important for lysosomal function. The glycolytic enzymes GAPDH and PGK1 were found associated with lysosomal vesicles, and NAD+ was required for ATP production around lysosomal vesicles. Thus, we conclude that NAD+ content affected by mitochondrial dysfunction is essential for lysosomal maintenance.

**Keywords** GAPDH; lysosome; mitochondria; NAD+; Nmnat3
**Subject Categories** Membranes & Trafficking; Organelles; Translation & Protein Quality
**The EMBO Journal (2021) 40: e105268**

## Introduction

Mitochondria regulate a multitude of different signaling and metabolic pathways and play an important role in the cell death program. Mitochondrial function has been considered to decline during aging together with oxidative phosphorylation (OXPHOS) dysfunction and morphological alterations (Shigenaga et al, 1994). Autophagy and autophagic flux are generally reduced in aged hearts, and mouse autophagy knockout models develop excessive cardiac dysfunction with accumulation of misfolded proteins and dysfunctional organelles (Fernandez et al, 2018). The mechanism by which autophagy diminishes during aging is extremely complex and remains elusive. In mitophagy, damaged mitochondria are surrounded by double membrane-bound autophagosomes, which are degraded via normal lysosomal degradation (Youle & Van Der Bliek, 2012; Kornfeld et al, 2015). In aging, lysosomal proteolytic activity is also decreased; however, it is unclear how mitochondrial dysfunction affects lysosomal function during aging.

Recently, mitochondrial dysfunction has been shown to have pleiotropic effects in multicellular organisms, affecting other organelles such as lysosomes. Mitochondrial respiratory failure affects lysosomal function and increases the accumulation of p62 and sphingomyelin. As a result, it disrupts the endolysosomal transport pathway and autophagy, resulting in mitochondrial dysfunction and impaired lysosomal accumulation (Baixauli et al, 2015). It has also been reported that mitochondrial respiratory chain deficiency inhibits lysosomal hydrolysis and lysosomal function, and the relationship between mitochondria and lysosomal function was described via various pathways (Demers-Lamarche et al, 2016; Fernandez-Mosquera et al, 2017; Fernandez-Mosquera et al, 2019; Deus et al, 2020). Thus, the crosstalk between mitochondria and lysosomal function has been revealed.

The mitochondrial protein, p32/complement component 1q binding protein (C1qbp), is an RNA and protein chaperone conserved among eukaryotes that is required for functional mitochondrial ribosome formation within mitochondria (Yagi et al, 2012). A p32 mutation was the suspected cause of a mitochondrial respiratory chain disorder in cardiomyopathy in four patients with a defect in mitochondrial energy metabolism (Kohda et al, 2016; Feichtinger et al, 2017). Mice lacking p32 in the central nervous system were also found to exhibit progressive oligodendrocyte loss, axonal degeneration, and white matter degeneration with vacuole formation in the midbrain and brainstem regions (Yagi et al, 2017). Cardiomyocyte-specific loss of mitochondrial p32/C1qbp (p32cKO) caused

1   Department of Clinical Chemistry and Laboratory Medicine, Kyushu University, Fukuoka, Japan
2   Department of Health Sciences, Graduate School of Medical Sciences, Kyushu University, Fukuoka, Japan
3   Department of Nutritional Sciences, Faculty of Health and Welfare, Seinan Jo Gakuin University, Kitakyushu, Japan
    *Corresponding author. Tel: +81 92 642 5750; Fax: +81 92 642 5772; E-mail: uchiumi@cclm.med.kyushu-u.ac.jp

cardiomyopathy owing to loss of mitochondrial translation, OXPHOS dysfunction, and structural impairment at 2 months of age, followed by death 10 months later (Saito *et al*, 2017). During starvation, p32 has been shown to be involved in ULK1 stability, suggesting that the p32-ULK1 autophagy axis is important in regulating stress response, cell survival, and mitochondrial homeostasis (Jiao *et al*, 2015). Hence, we decided to investigate how mitochondrial translational defects interfere with autophagy function, especially lysosomal function.

Lysosomes, the degradative organelles of the endocytic and autophagic pathways, function at acidic pH. Lysosome acidification is achieved by vacuolar ATPase (V-ATPase), which allows protons to enter the lysosome lumen in an ATP-dependent manner (Ohkuma *et al*, 1982; Breton & Brown, 2013). However, the mechanism by which ATP is supplied to V-ATPase in lysosomes is not well understood. Lysosome and autophagy dysfunction are associated with age-related neurodegenerative diseases as a result of neurons accumulating damaged organelles such as long-lived or misfolded proteins and mitochondria (Menzies *et al*, 2015).

The relationship between lysosomal function and nicotinamide adenine dinucleotide (NAD$^+$) in Tfam-deficient T cells has been reported (Baixauli *et al*, 2015). NAD is an essential cofactor that mediates several biological processes such as DNA repair, gene expression, and enzyme metabolism (Canto *et al*, 2015). It has been suggested that changes in the balance between NAD synthesis and consumption reduce NAD levels during aging. Decreased NAD level is also involved in aging-related diseases such as metabolic diseases, cancer, and neurodegenerative diseases (Fang *et al*, 2017). Nicotinamide mononucleotide adenylyl transferases (Nmnats) are a family of highly conserved proteins essential for cell homeostasis. Nmnat deficiency has been associated with a variety of human diseases that have prominent consequences for neural tissues, highlighting the importance of neuronal maintenance (Conforti *et al*, 2007). Recently, it has been suggested that NAD-related metabolites have some unique biological functions in the regulation of disease and longevity (Yaku *et al*, 2018).

In the present study, we examined how mitochondrial translation deficiency induces loss of lysosomal function and provided new insight into the essential nature of NAD$^+$ in normal lysosomal function.

## Results

### Cardiomyocyte-specific knockout of p32 shows impaired autophagy in the heart

Cardiomyocyte-specific loss of mitochondrial p32/C1qbp causes cardiomyopathy owing to loss of mitochondrial translation and impaired mitochondrial structure (Saito *et al*, 2017). The loss of p32 may result in more defective mitochondria, and therefore, the removal of the damaged mitochondria would be essential for cell homeostasis. Electron microscopy of heart sections from p32cKO mice at 6 months revealed that some mitochondria were larger and showed more internal collapse, compared with the control (Figs 1A and EV1A). This observation led us to hypothesize that removal of the damaged mitochondria is decreased in the heart of p32cKO mice. It was reported that p32 is involved in autophagy by ULK axis

(Jiao *et al*, 2015), we first investigated the expression of autophagic molecule. LC3, p62, and ubiquitin are central autophagy-related proteins involved in autophagic flux. The expression of LC3-II and p62 was significantly increased in the heart of p32cKO mice compared with wild-type (WT) hearts (Fig 1B). Immunostaining of LC3, p62, and ubiquitin was stronger in p32cKO mouse heart sections compared with the WT, which was consistent with the Western blot results (Figs 1C, and EV1B and C). Additionally, a ring-like staining pattern was observed for LC3, p62, and ubiquitin in the heart of p32cKO mice (Fig 1C). Furthermore, we observed co-localization of p62 and ubiquitin and co-localization of p62 and the mitochondrial protein, PDH, in the p32cKO heart (Fig 1D and E), indicating that mitochondria are surrounded by autophagy-related protein. The size of the p62 and LC3 ring staining pattern was similar to that of the giant mitochondria determined by electron microscopy analysis (Fig 1A), suggesting that LC3-positive ring sequestrate damaged mitochondria in the p32cKO heart. When autophagy is triggered, mitochondria elongate *in vitro* and *in vivo* (Rambold *et al*, 2011), suggesting that mitochondrial translation deficiency heart shows the elongated mitochondria. Consistent with these results, increased ubiquitinated protein levels were detected in the p32cKO heart (Fig 1F).

Next, we examined the phosphorylation of the autophagy-related proteins, p62 and ULK1, and the progress of autophagy. Ser351 phosphorylation of p62 was not altered in p32cKO compared with WT samples, but Ser403 phosphorylation, which is induced by ULK1, was clearly increased (Fig 1G). Furthermore, the phosphorylation of ULK1 Ser757 in the p32cKO heart was increased fourfold compared with WT samples (Fig EV1D), which is consistent with the autophagy initiation process by LC3-II ring formation. It has been suggested that AMPK activated autophagy via two independent mechanisms: suppression of mammalian target of rapamycin complex 1 (mTORC1) activity and direct control of ULK1 phosphorylation (Zhao & Klionsky, 2011). Previously, we have observed AMPKα phosphorylation in p32cKO hearts (Saito *et al*, 2017), which seems to initiate autophagy, but the subsequent steps are suppressed in the heart of p32cKO mice. The expression of the autophagy-related genes, *Gabarapl1*, *Lamp2*, and *Atg4b*, was increased, whereas that of the muscle-specific ubiquitin ligase, *Atrogin1*, was decreased in the heart of p32cKO mice compared with that in the heart of WT mice (Fig EV1E). It has been reported that *Atrogin1* deficiency promotes cardiomyopathy and premature death via impaired autophagy (Zaglia *et al*, 2014). Consequently, the autophagy-related molecules are gathered around the damaged mitochondria as a result of autophagy suppression.

### Lysosomal morphology is impaired in the heart of p32cKO mice

Next, we examined the possibility that the suppression of the autophagic degradation was due to lysosomal abnormality. We performed ultrastructural analysis of lysosomes using electron microscopy. In the WT heart, the lysosomes were small and in high density, while the lysosomes in the p32cKO heart at 6 months were large and in low density (Fig 2A). The lysosomes in the p32cKO heart were significantly longer in diameter than in the WT heart. These structural changes suggest the presence of lysosomal abnormalities in p32cKO hearts (Fig 2A). Lipofuscin originates from intralysosomal

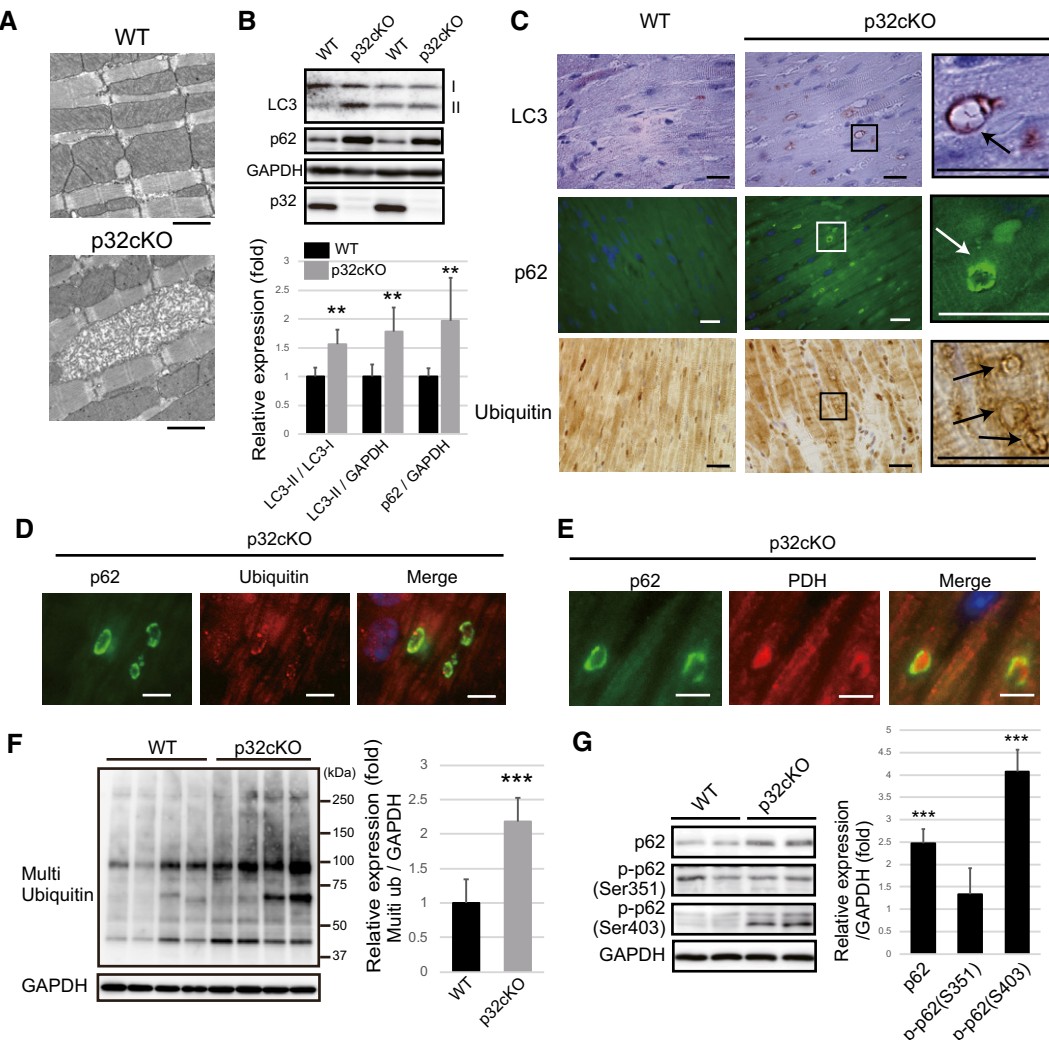

**Figure 1.  Autophagy is suppressed in the heart of p32cKO mice.**

A   Electron microscopy of abnormal mitochondria in p32cKO hearts of 6-month-old mice. These two figures are also presented in Fig EV1A along with other EM figures. Scale bars, 1 μm.

B   Western blot analysis of autophagy marker proteins (LC3-II and p62) in 6 months old WT and p32cKO hearts. The bands of LC3-II and p62 are increased in the p32cKO heart. GAPDH was used as an internal control. Quantification is shown on the bottom. These values are presented as mean ± SD (*n* = 6 mice per group). Student's t-test was performed on WT vs. p32cKO, **P < 0.01.

C   Immunostaining of LC3, p62, and multi-ubiquitin in heart tissues of 6 months old. The magnified image (right panel) shows the ring-like staining pattern in the p32cKO heart (indicated arrows). Scale bar, 20 μm. The panel related to LC3 immunostaining in p32cKO is also presented in Fig EV1B (top left) along with other immunostaining pictures.

D   In the 6-month-old p32cKO heart, p62 and multi-ubiquitin staining patterns were co-localized. Scale bar, 5 μm.

E   In the 6-month-old p32cKO heart, the staining patterns of p62 and the mitochondrial marker protein, pyruvate dehydrogenase (PDH), were co-localized. Scale bar, 5 μm.

F   Western blot analysis of ubiquitinated proteins using an anti-multi-ubiquitin antibody in the 6-month-old WT and p32cKO heart (*n* = 4 mice per group). Error bars are presented as mean ± SD. Student's *t*-test was performed on WT vs. p32cKO, ***P < 0.005.

G   Western blot analysis of phosphorylated p62 (Se351 and Ser403) and non-phosphorylated p62 in the 6-month-old WT and p32cKO heart. Relative expression of phosphorylated p62 is shown on the right. Error bars are presented as mean ± SD. Student's *t*-test was performed on eight WT mice vs. seven p32cKO mice, ***P < 0.005.

Source data are available online for this figure.

components that have become oxidized outside or inside the lysosomal compartment during autolysosome formation (Stroikin *et al*, 2007). Broad-spectrum autofluorescence is a characteristic property of lipofuscin. In the p32cKO heart, the number of autofluorescing particles increased around the nuclei over time, whereas in the WT heart it did not (Figs 2B and EV1F). Collectively, lysosomes in p32cKO hearts seemed morphologically impaired.

To more specifically elucidate the mechanism, we examined the localization and size of lysosomes by immunostaining Lamp2 in the presence of TrueBlack to quench autofluorescence. Many large dots

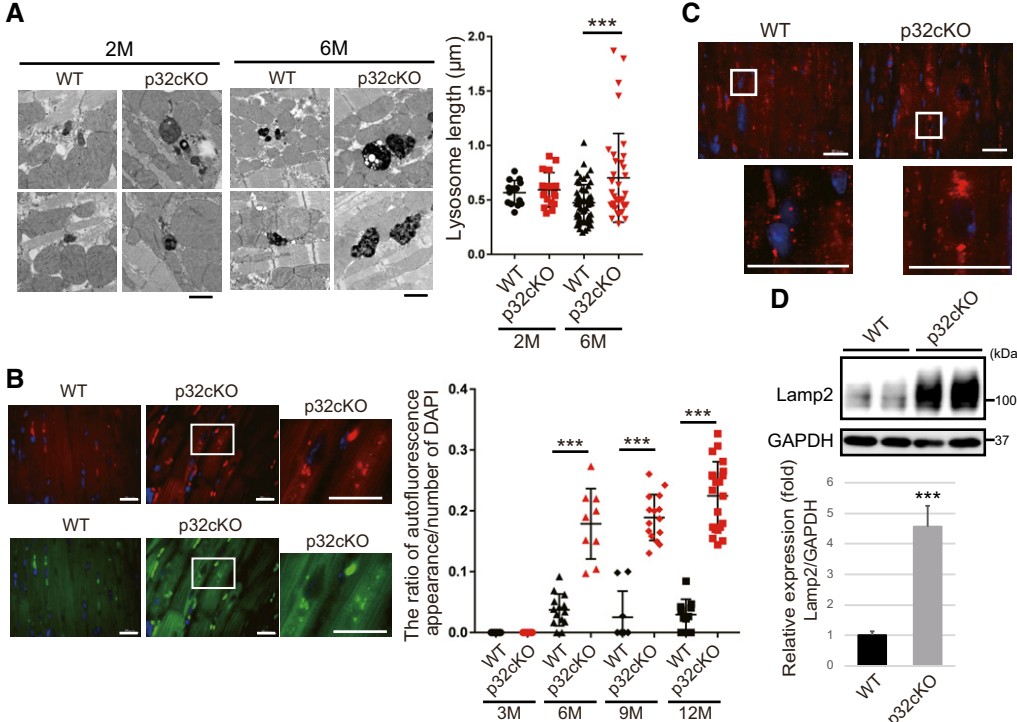

**Figure 2. Cardiomyocyte-specific p32cKO causes lysosomal enlargement in mice.**

A   Electron micrographs showing lysosomes in cardiac myofibrils of 2 and 6 months old mice. Lysosome size and density were different in WT (black dot) and p32cKO (red dot) heart tissues. Scale bars, 1 μm. The quantification of the average lysosomal length is presented in the plot in the right panel as mean ± SEM (2-month-old mice, $n$ = 14–16; 6-month-old mice, $n$ = 35–60). Statistical significance was assessed by Student's $t$-test, ***$P$ < 0.005.

B   Autofluorescence showing lipofuscin localization around the nucleus in the 6 months old p32cKO heart. Tissues were excited at a wavelength of 540 (upper panel) or 470 (lower panel), and emission spectra were collected with a confocal microscope at wavelengths (band path) of 580–630 nm (upper panel) or 510–560 nm (lower panel). Enlarged view of the white squares is shown right. Scale bars, 20 μm. The quantification of the ratio of autofluorescence of DAPI staining is presented in the plot in the right panel as mean ± SEM (3-month-old mice, $n$ = 11; 6-month-old mice, $n$ = 9–14, 9-month-old mice, $n$ = 9–15, 12-month-old mice, $n$ = 11–21). Statistical significance was assessed by Student's $t$-test, ***$P$ < 0.005.

C   Immunostaining of Lamp2 in the heart using TrueBlack™ to quench lipofuscin autofluorescence in the 6-month-old WT and p32cKO heart. Enlarged view of the white squares is shown below. Scale bars, 20 μm.

D   Western blot analysis of Lamp2 in the 6 months old WT and p32cKO hearts. Quantification is shown in the lower panel ($n$ = 4 mice per group). GAPDH was used as an internal control. Error bars are presented as mean ± SD. Student's $t$-test was performed on WT vs. p32cKO, ***$P$ < 0.005.

Source data are available online for this figure.

were observed around the nuclei in the p32cKO heart (Fig 2C). Furthermore, Lamp2 expression in the p32cKO heart was significantly higher than that in the WT heart (Figs 2D and EV1G). Taken together, we speculate that the degradation mechanism by autophagy was stopped prematurely and impaired lysosomal morphology might be involved in autophagic dysfunction.

**p32-deficient heart shows reduced NAD synthesis**

To elucidate the molecular mechanism of impaired lysosomal morphology in the p32cKO heart, we performed metabolome analysis related to energy sources such as NAD by LC-MS/MS of the hearts at 6 months (Table EV1). The amounts of NAD$^+$ and NADP$^+$ were significantly reduced in the p32cKO heart compared with the WT levels (Fig 3A). However, the amounts of NADH, nicotinamide, and nicotinic acid adenine dinucleotide (NAAD) did not change. The decrease in NAD$^+$ content may be due to a decrease in NAD$^+$ synthesis or an increase in NAD$^+$ consumption. Accordingly, we

examined the gene expression of NAD$^+$ synthesis enzymes. The gene expression of enzymes in the salvage pathway such as *Nmnat1–3* and *Nampt* was decreased in the p32cKO heart compared with that in the WT heart (Fig 3B). We also confirmed that Nmnat3 protein was decreased in the p32cKO heart (Fig 3C), suggesting that the reduced NAD$^+$ content was due to reduced expression of NAD$^+$ synthesis genes.

We next examined the NAD$^+$ content and Nmnat activity in the cytosol of p32cKO heart tissue using a colorimetric assay. NAD$^+$ content in the p32cKO heart was lower than that in the WT heart without nicotinamide mononucleotide (NMN) addition, which is consistent with the LC-MS-based metabolite analysis (Fig 3D, left). The Nmnat activity, assessed by NAD$^+$ formation after the addition of NMN, was also reduced in the p32cKO heart tissue compared with that in the WT heart (Fig 3D, right), indicating that the level of NAD$^+$ synthesis was reduced at least in part because of a decrease in the Nmnat activity resulting from the lower *Nmnat* gene expression in the p32cKO heart.

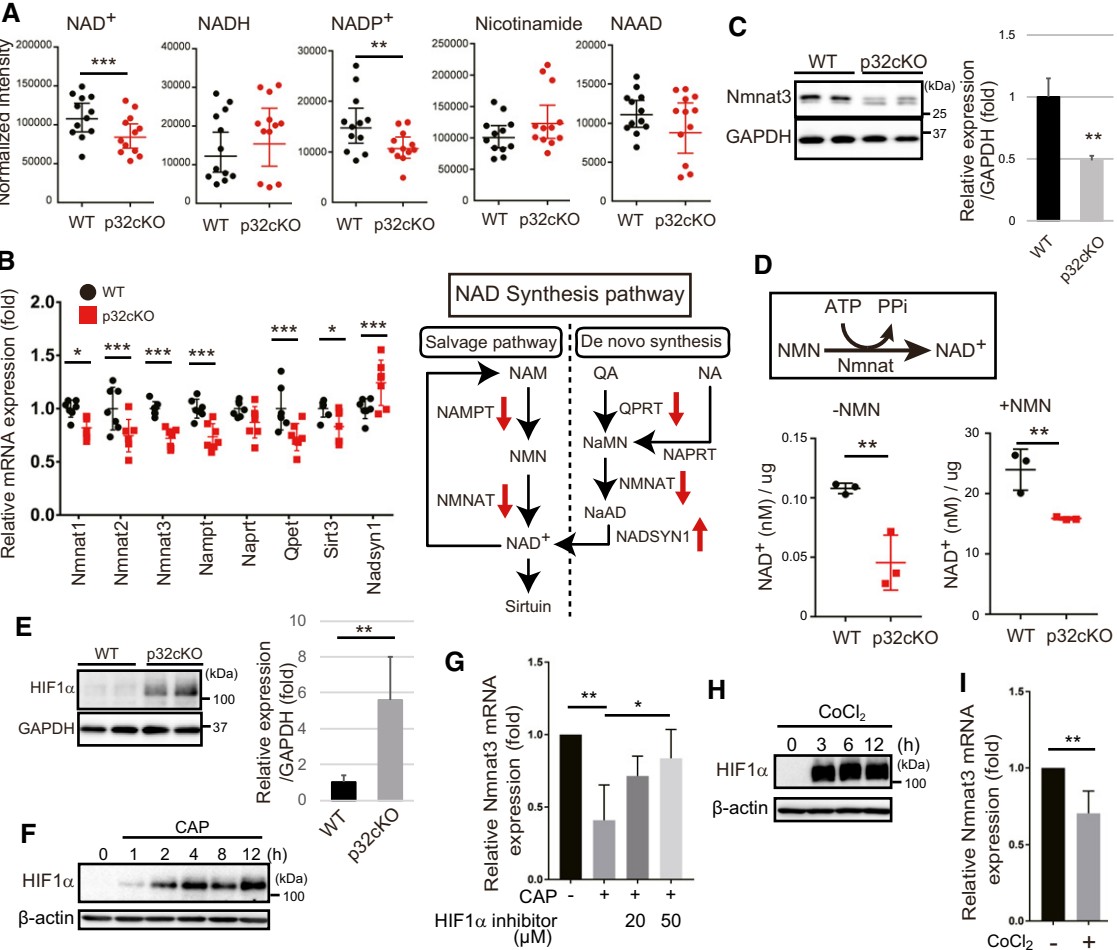

**Figure 3. Reduced NAD synthesis gene expression and reduced NAD levels in the p32cKO heart.**

A  LC-MS/MS metabolomic analysis of NAD$^+$, NADH, NADP$^+$, nicotinamide, and NAAD in the 6-month-old WT and p32cKO hearts. NAD$^+$ and NADP$^+$ showed significantly different levels between WT and p32cKO hearts ($n = 12$). Error bars are presented as mean $\pm$ SD. Student's *t*-test was performed on WT vs. p32cKO, **$P < 0.001$, ***$P < 0.0002$.

B  Real-time PCR analysis of RNA expression of NAD-synthesizing enzymes in the 6 months old WT and p32cKO hearts. Error bars are presented as mean $\pm$ SD. Student's *t*-test was performed on WT mice vs. p32cKO mice ($n = 6$), ***$P < 0.005$, *$P < 0.05$. The right panel shows the NAD synthesis pathway and mRNA expression levels are indicated by red arrows. NAM: nicotinamide, NA: nicotinic acid, QA: quinolic acid, NMN: nicotinamide mononucleotide.

C  Reduced Nmnat3 expression in 9-month-old p32cKO heart. Quantification is shown on the right ($n = 4$ mice per group). Error bars are presented as mean $\pm$ SD. Student's *t*-test was performed on WT vs. p32cKO, **$P < 0.01$.

D  NAD$^+$ levels without or with NMN addition in 9-month-old WT and p32cKO heart lysates ($n = 3$). Error bars are presented as mean $\pm$ SD. Student's *t*-test was performed on WT vs. p32cKO, **$P < 0.002$.

E  Western blot analysis of HIF1α in 9-month-old WT and p32cKO heart. Quantifications is shown on the right ($n = 4$ mice per group). Error bars are presented as mean $\pm$ SD. Student's *t*-test was performed on WT vs. p32cKO, **$P < 0.01$. GAPDH was used as an internal control.

F  Immunoblot analysis of HIF1α; the expression in 3T3-L1 cells was increased after treatment with 1 mM chloramphenicol (CAP), which inhibits mitochondrial translation. β-actin was used as an internal control. One representative experiment out of three shown.

G  *Nmnat3* mRNA expression in 3T3-L1 cells after 1 mM CAP treatment for 72 h. The HIF1α inhibitor (20 or 50 μM) was added 2 h before the CAP treatment. Error bars are presented as mean $\pm$ SD of three independent experiments. Student's t-test was performed on WT cells vs. WT cells treated CAP and (or) HIF1α inhibitor, **$P < 0.01$, *$P < 0.05$.

H  Immunoblot analysis of HIF1α after treatment with 150 μM CoCl$_2$ in 3T3-L1 cells, which stabilized HIF1α. β-actin was used as an internal control.

I  *Nmnat3* mRNA expression in 3T3-L1 cells after 150 μM CoCl$_2$ treatment for 72 h ($n = 3$). Error bars are presented as mean $\pm$ SD of three independent experiments. Statistical significance was assessed by Student's *t*-test, **$P < 0.01$.

Source data are available online for this figure.

To elucidate which Nmnat isozyme is responsible for this process, we examined the expression and localization of cytoplasmic Nmnat2 and Nmnat3 in cardiac tissue, because Nmnat1 is mainly localized in the nucleus (Fig EV2A). We isolated the membrane and cytosol fraction from WT heart (Fig EV2B). Consistent with a previous report that Nmnat2 is mainly expressed in the brain (Ali *et al*, 2013), no protein expression was detected in the heart tissue (Fig EV2C). Nmnat3 was detected

in the p32cKO heart (Fig 3C) and was thought to localize in mitochondria. However, recently its localization in the cytosol has been reported (Yamamoto *et al*, 2016). In fact, abundant Nmnat3 was also found in the cytosol of the heart (Fig EV2C). These results suggest that Nmnat3 is expressed in the cytosol and the membrane fraction of the heart and contributes to NAD⁺ synthesis in the cytosol.

### Mitochondrial translation deficiency induces HIF1α to inhibit *Nmnat3* expression

We investigated the mechanism of *Nmnat3* downregulation in the p32cKO heart. We focused on the transcription factor, HIF1α, which is involved in *Nmnat3* expression in *Drosophila* (Ali *et al*, 2011). We found that HIF1α was significantly upregulated in the p32cKO heart compared with the WT heart (Fig 3E). Because p32 is involved in mitochondrial translation, we used antibiotics such as chloramphenicol (CAP) to inhibit mitochondrial translation. CAP induced HIF1α expression in mouse 3T3-L1 cells in a time-dependent manner (Fig 3F). Moreover, CAP treatment reduced *Nmnat3* expression (Fig 3G). We also observed that CoCl₂ treatment, which stably induces HIF1α expression, suppressed *Nmnat3* gene expression (Fig 3H and I). A chromatin immunoprecipitation (ChIP) database analysis (ChIP-Atlas: http://chip-atlas.org/) showed that HIF1α can associate with promoter regions of the *Nampt*, *Nmnat1–3*, and *Sirt3* genes in several cell lines (Fig EV2D). In contrast, a HIF1α inhibitor suppressed the CAP inhibitory effect on *Nmnat3* expression (Fig 3G), suggesting that mitochondrial translation inhibition induced HIF1α expression, leading to suppression of *Nmnat3* expression.

### NMN rescues lysosomal acidification

Our findings prompted us to examine the link between decreased NAD⁺ and lysosomal morphological changes in a heart with mitochondrial translation deficiency. To examine lysosomal acidification, we used two fluorescently tagged probes: the pH-sensitive Oregon green 488–dextran and the pH-insensitive tetramethyl rhodamine-dextran. Oregon green 488 has a pKa of 4.7, which is suitable for measuring the acidic pH of the lysosomal lumen. The emission was separately determined in individual lysosomes, and the fluorescence ratio was measured, enabling changes in lysosomal pH to be monitored (more acidic shows lower green/red ratio, whereas less acidic shows higher green/red) (Johnson *et al*, 2016). We observed less acidification in p32KO MEFs compared with WT cells, suggesting that lysosomal acidification was reduced in p32KO MEFs (Fig 4A). Addition of NMN to p32KO MEFs restored the lysosomal acidification (Fig 4A). Furthermore, we observed less acidification upon CAP treatment in WT MEFs, suggesting that mitochondrial translation deficiency affects lysosomal acidification. Accordingly, treatment of MEFs with the V-ATPase inhibitor, bafilomycin A1, resulted in clear decreased lysosomal acidification (Figs 4A and EV3A). These results imply an association between NAD⁺ synthesis and lysosomal acidification in p32-deficient cells.

Next, we examined whether forced expression of NAD⁺ synthesis enzymes such as Nmnat2 and Nmnat3 restores lysosomal acidification in p32KO MEFs. Indeed, the expression of Nmnat2 in p32KO MEFs led to restoration of lysosomal acidification (Fig EV3B and C),

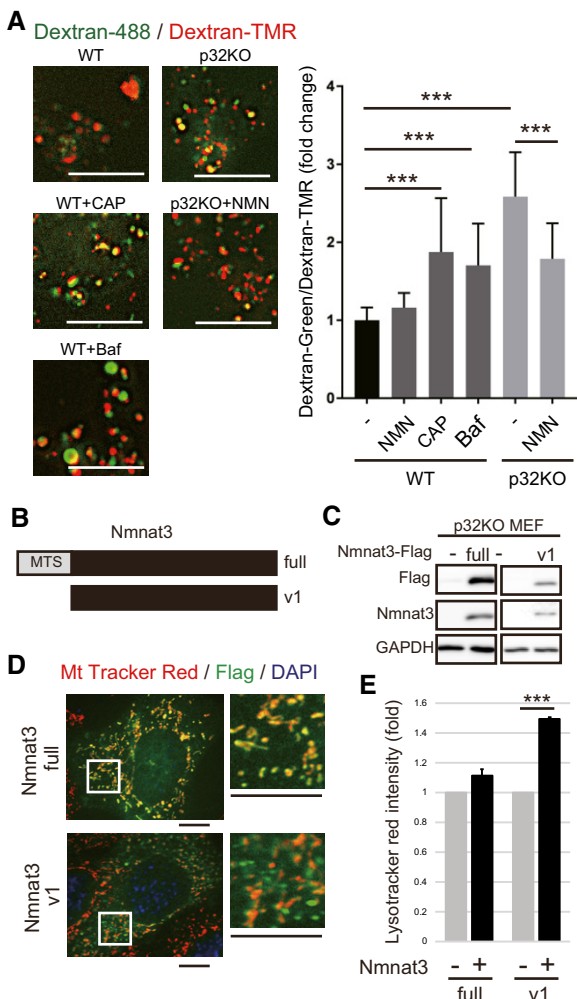

**Figure 4. NMN addition and Nmnat overexpression rescue lysosomal function.**

A Lysosomal acidification is impaired in p32KO MEFs. Representative images of WT and p32KO MEFs, stained with dextran-Oregon Green (488) and dextran-TMRM. Scale bar, 5 μm. Addition of 1 mM NMN for 48 h to p32KO MEFs rescued lysosomal acidification, and 1 mM CAP treatment of WT MEFs for 48 h decreased lysosomal acidification. Dextran-Oregon Green is quenched under acidic pH, and thus, an increased green:red ratio denotes impaired lysosomal acidification. The average ± SEM of the green:red ratio in at least 30 cells in two independent experiments is presented in the plot (normalized to WT green/red = 1). Statistical significance was assessed by Student's *t*-test, *P < 0.05, ***P < 0.001.

B Two Nmnat3 constructs: Nmnat3 (full) with a mitochondrial-targeting sequence (MTS), and Nmnat3(v1) without MTS.

C Both these constructs were effectively expressed in p32KO MEFs. After transfection, lysates were subjected to Western blot analysis with anti-Flag and anti-Nmnat3 antibodies. GAPDH was used as an internal control. One representative experiment out of three shown.

D Nmnat3 (full) was localized in mitochondria, but Nmnat3(v1) did not co-stain with MitoTracker red in p32KO MEFs. Scale bars, 10 μm. The magnified image is shown on the right.

E Nmnat3(v1) overexpression in p32KO MEFs rescued lysosomal function. LysoTracker Red-stained p32KO MEFs were quantified by flow cytometry. Error bars are presented as mean ± SEM of 4 independent experiments. Statistical significance was assessed by Student's *t*-test, ***P < 0.001.

Source data are available online for this figure.

suggesting that NAD$^+$ synthesis is involved in lysosomal acidification. However, as mentioned before, Nmnat3 is expressed in the heart tissue (Fig 3C), but Nmnat2 is not (Fig EV2C). The *Nmnat3* gene has two splice variants, one has a mitochondrial-targeting sequence (MTS) (full) and the other does not (v1) (Fig 4B and C). We transfected plasmids expressing *Nmnat3* cDNA with or without MTS into p32KO MEFs to examine the expression and localization of Nmnat3 in relation to the effects on lysosomal acidification. We observed that Nmnat3 (full) was localized in mitochondria, whereas Nmnat3 (v1) was localized in cytoplasmic compartments other than mitochondria (Fig 4D). Nmnat3 (v1) was able to rescue the lysosomal acidification in p32KO MEFs, while the mitochondria-localized Nmnat3 (full) had no effect (Figs 4E and EV3D and E). These findings suggest that cytosolic NAD$^+$ is involved in lysosomal acidification and mitochondrial p32 may regulate lysosomal function via cytosolic NAD$^+$ synthesis.

## FK866 inhibits lysosomal and autolysosomal function

To test the involvement of NAD$^+$, we examined whether NAD$^+$ synthesis inhibition reduces lysosomal and autolysosomal function. Nampt is a rate-limiting enzyme in NAD$^+$ synthesis. Treatment of WT MEFs with FK866, a Nampt inhibitor, depleted intracellular NAD$^+$ and NADH (Fig EV4A) and reduced lysosomal acidification (Fig 5A). Because NMN is a product of the enzymatic reaction of Nampt, NMN was anticipated to restore NAD$^+$ levels and rescue lysosomal function in the presence of FK866. Addition of NMN to FK866-treated cells restored the lysosomal acidification (Fig 5A) as assessed by dextran-488 and dextran-TMR, indicating that lysosomal function requires NAD$^+$ synthesis. We also estimated lysosomal acidification using another probe, LysoSensor DND-160. The intralysosomal pH measured by this probe increased upon FK866 addition, and this lysosomal pH increase was rescued by NMN (Fig 5B).

To assess whether the increased pH affects lysosomal proteolytic activity, we incubated the FK866-treated MEFs with bovine serum albumin labeled with a green BODIPY dye. DQ green BSA is a high-molecular-weight DQ that contains a large amount of fluorophore and has a quenching effect. DQ Green BSA accumulated in the lysosome via endocytosis and is hydrolyzed by lysosomal protease to small peptides which does not quench fluorescence (Marwaha & Sharma, 2017). Moreover, we can assess the activity of lysosomal proteases by measuring this rate of increase in fluorescence.

Compared with the control, FK866-treated WT MEFs had a lower rate of DQ Green BSA hydrolysis, suggesting a decrease in the activity of lysosomal proteases, which indicates that lysosomal function requires NAD$^+$ synthesis (Fig 5C). CAP-treated MEFs and p32KO MEFs also showed reduced lysosomal degradation activity in a time-dependent manner, suggesting that mitochondrial translation deficiency affected the lysosomal degradation activity (Figs 5D and EV4B).

We next examined the formation of autophagic vesicles (autophagosomes and autolysosomes) after FK866-induced NAD$^+$ depletion. In steady-state cells, p62 and Lamp2 did not co-localize (Fig 5E, top panel), but bafilomycin A treatment induced co-localization due to the suppression of p62 degradation because of lysosomal dysfunction (Fig 5E, bottom panel). We also observed that p62 and Lamp2 partially co-localized after addition of FK866 (Fig 5E,

central panel), suggesting that inhibition of NAD$^+$ synthesis decreased lysosomal function and the formed autolysosome could not efficiently degrade its contents.

DALGreen fluorescence is enhanced at an acidic pH and is suitable for monitoring the autophagy degradation stage, also known as the autolysosome stage. In contrast, DAPRed has a pH-independent fluorescence profile and remains fluorescent with almost constant intensity throughout the process of autophagy (Fig EV4C). We investigated the formation of nutrient-starved autophagosomes (DAPRed) and autolysosome vesicles (DALGreen) after treatment with FK866 or bafilomycin A. After bafilomycin A treatment, an increased number of fusion of DAPRed and DALGreen staining was observed, suggesting a reduction in autophagic degradation within autolysosomes (Fig 5F). We also found increased number of fusion of DAPRed and DALGreen staining after FK866 treatment, suggesting that reduced NAD$^+$ levels inhibited autophagic degradation because of lysosomal dysfunction followed by accumulation of both autophagosomes and autolysosomes (Fig 5F).

Seeking more evidence, we examined the expression of lysosomal proteins and autophagy proteins after FK866 treatment. The expression of Lamp2, p62, and LC3-II increased after FK866 treatment in a time-dependent manner (Fig EV4D). These results further indicate that inhibition of NAD$^+$ synthesis impairs lysosomal function and consequently suppresses degradation by autolysosomes.

## Lysosomal fractions contain GAPDH, PGK1, and NAD$^+$, which are linked to ATP production around lysosomes

To investigate how NAD$^+$ levels are involved in lysosomal acidification, we hypothesized that ATP, which is necessary for lysosomal acidification, is generated around lysosomal vesicles to be efficiently utilized by V-ATPase on the lysosomal membrane. GAPDH is mainly localized in the cytosol, but it has also been reported that it is localized in some intracellular compartments such as the membrane and vesicles due to partial palmitoylation (Tristan *et al*, 2011). In the glycolytic pathway, GAPDH is in a complex with 3-phosphoglycerate kinase (PGK) (Srivastava & Bernhard, 1986) and ATP is generated by this complex using NAD$^+$ as a substrate.

We first isolated the lysosomal fraction, and then examined whether GAPDH and PGK1 are localized in this fraction. To purify lysosomal fractions, the membrane fraction was washed twice to remove the free cytosolic proteins and then adjusted with a discontinuous OptiPrep™ density gradient (Fig 6A). After ultracentrifugation, five individual fractions were collected from the top of the gradient and then analyzed for the presence of characteristic organelle marker proteins (Fig 6B).

After two washes and centrifugations, the membrane fraction was concentrated. Cytosolic proteins were hardly detected in the wash2, suggesting the input membrane fraction contains little soluble cytosolic protein contamination (Fig EV5A and B). After further separation of the membrane fraction by the OptiPrep gradient centrifugation, Lamp2 and V-ATPase were enriched in fractions 2 and 3 (lysosomal fraction). Consistent with a part of the glycolytic enzyme Hexokinase 2 (HK2) is known to be associated to mitochondria, HK2 was mainly detected in the mitochondrial fraction where mitochondrial voltage-dependent anion carrier (VDAC) is enriched (fractions 4 and 5). While HK2 and cytosolic HSP70 were very weakly found in fraction 3, they were scarcely found in fraction 2,

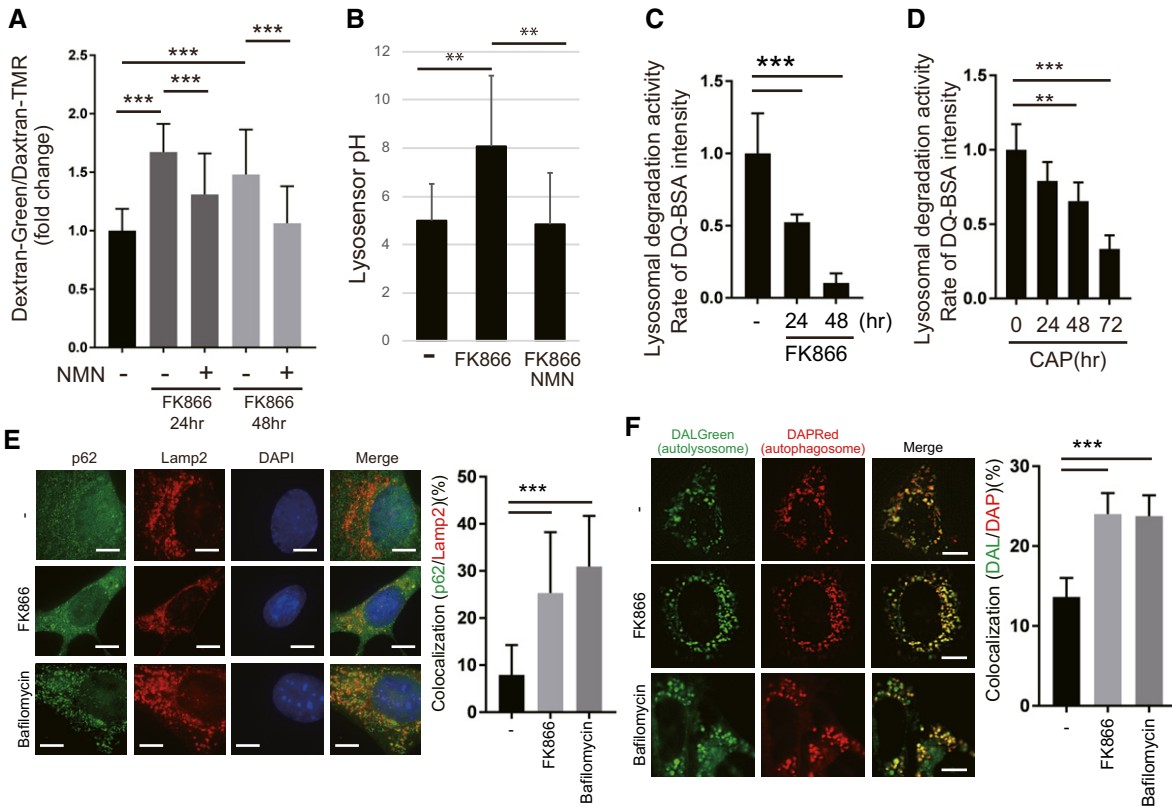

**Figure 5. The Nampt inhibitor, FK866, decreases lysosomal acidification and suppresses autolysosome function.**

A WT MEFs were stained with dextran-Oregon Green and dextran-TMRM to examine lysosomal acidification after 10 nM FK866 treatment. FK866 decreased lysosomal acidification, which was rescued by 1 mM NMN pretreatment. The stained MEFs were quantified using a microscope (BZ-X800, KEYENCE). Error bars are presented as mean ± SEM of three independent experiments. Statistical significance was assessed by one-way ANOVA, ***$P < 0.001$.

B LysoSensor™ Yellow/Blue DND-160 was used to detect lysosomal pH function in WT MEFs. Lysosomal acidification was decreased by 10 nM FK866 and rescued by 1 mM NMN. Error bars are presented as mean ± SEM of three independent experiments. Statistical significance was assessed by one-way ANOVA, **$P < 0.01$.

C Lysosomal proteolytic capacity was decreased in FK866-treated WT MEFs. DQ-BSA is bovine serum albumin labeled with a green fluorophore. It is taken up by endocytosis and delivered to the lysosomes. As a monomer, the fluorophore is too concentrated and the signal is quenched. Because DQ-BSA is degraded by the lysosomal proteases, it releases monomers, which emit fluorescence. The rate of DQ-BSA hydrolysis (increase in fluorescence units per cell) is a function of lysosomal proteases and was measured using a plate reader over 1 h. The linear range was used to determine the rate. The mean ± SEM rate for each condition is presented in the plot. These values are of a representative experiment. Three independent experiments were performed. Statistical significance was assessed by one-way ANOVA, ***$P < 0.001$.

D Lysosomal proteolytic capacity was decreased after 1 mM CAP treatment of WT MEFs in a time-dependent manner, as shown by DQ-BSA. Error bars are presented as mean ± SEM of three independent experiments. Statistical significance was assessed by one-way ANOVA, **$P < 0.01$, ***$P < 0.001$.

E To examine autolysosome formation, we performed immunofluorescence with Lamp2 and p62 antibodies. Lamp2 and p62 were co-localized after 10 nM FK866 or 50 nM bafilomycin A treatment in WT MEF cells. Scale bars, 10 μm. Quantification is shown on the right. Error bars are presented as mean ± SEM of three independent experiments. Statistical significance was assessed by one-way ANOVA, ***$P < 0.001$.

F To examine autophagosome and autolysosome formation, we used DAPRed, which indicates autophagosomes, and DALGreen, which indicates autolysosomes. After starvation, co-localization of both staining agents was observed upon FK866 or bafilomycin A treatment in WT MEF cells. Scale bars, 10 μm. Quantification is shown on the right. Error bars are presented as mean ± SEM of three independent experiments. Statistical significance was assessed by one-way ANOVA, ***$P < 0.001$.

suggesting fraction 2 contains less mitochondrial and cytosolic contamination than fraction 3. Therefore, we focused on fraction 2 though Lamp2 is better enriched in fraction 3. We found that PGK1 was most enriched in fractions 2. GAPDH was present in all fractions because GAPDH is localized in specific subcellular compartments including fusion vesicles by palmitoylation. This separation study strongly suggests that PGK1 and GAPDH are physically associated with lysosome.

We next evaluated whether GAPDH and PGK1 are in a complex in the cell by co-immunoprecipitation experiments. MEF lysates were incubated with anti-PGK1 or anti-GAPDH antibodies; the resulting immunoprecipitants were subjected to Western blotting with anti-GAPDH or anti-PGK1 antibodies, respectively. GAPDH or PGK1 was detected in the immunoprecipitation elution, indicating that GAPDH and PGK1 may exist in a complex (Fig 6C). To determine the nature of the interaction of GAPDH and PGK1 with lysosomal vesicles, lysosomal vesicles were treated with various concentrations of NaCl. PGK1 and GAPDH dissociated from lysosomal membranes with increasing salt concentrations (Fig 6D), suggesting that GAPDH and PGK1 are associated with lysosomal vesicle membranes peripherally by ionic bonds. At 0.8 M NaCl, GAPDH and PGK1 were found to remain associated with lysosomal

vesicle membranes, suggesting that GAPDH and PGK1 are strongly associated with lysosomal vesicle membranes.

These observations strongly suggest that ATP is produced by the lysosome-bound GAPDH/PGK1 complex, which should more effectively support vesicular proton uptake than free cytosolic ATP. The activity of GAPDH and PGK1 is energetically coupled and uses GAP, NAD, Pi, and ADP to yield 3-PG, ATP, NADH, and H$^+$ (Weber & Bernhard, 1982; Srivastava & Bernhard, 1986) (Fig 6E). We measured ATP production by incubating lysosomal vesicle with GAP, ADP, Pi, and NAD, and significant GAP-dependent ATP production was observed under these conditions (Fig 6F). These results suggested that glycolytic protein such as GAPDH and PGK1 is enzymatically active in the lysosomal fraction. The GAP-dependent ATP production required full GAPDH catalytic function

because it was completely blocked by iodoacetate (IA). These results demonstrated that lysosomal vesicles harbor the glycolytic machinery that produces ATP by consuming NAD$^+$ in an autonomous manner.

To further verify the idea described above, we isolated lysosomal vesicles by two different immunoprecipitation methods. First, we created 3T3-L1 cells that express the transmembrane protein 192 (TMEM192) fused to three tandem HA epitopes. The lysosomal fractions were then isolated from these cells by immunoprecipitation (LysoIP) using HA-bound beads (Abu-Remaileh *et al*, 2017) (Fig EV5C and D). We confirmed that TMEM192-HA was localized to lysosomes with Lamp2 staining (Fig EV5E). Lysosomal purification by this method showed that the lysosomal marker, Lamp2, was enriched and that GAPDH and PGK1 were present in this fraction;

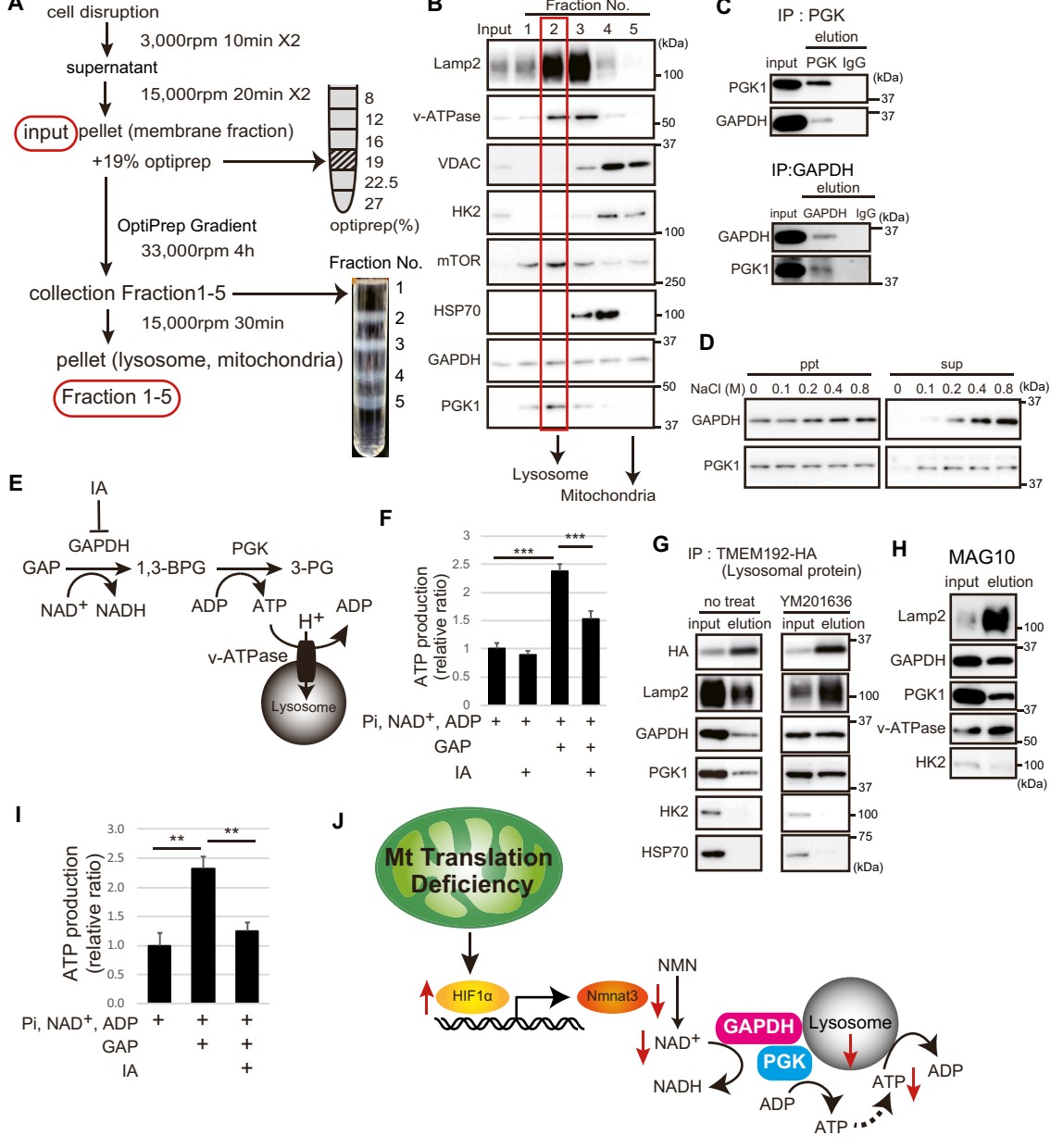

**Figure 6.**

◀

**Figure 6. The lysosomal fraction contains GAPDH and PGK1 and contributes to ATP production by NAD$^+$.**

A   Schema of purification of lysosomes by centrifugation. We applied a low osmotic medium for density gradient centrifugation. After primary organelle enrichment by differential centrifugation from WT MEFs, the subcellular compartments were separated on a discontinuous iodixanol gradient. After membrane pellets were enriched (input), each organelle was purified by an OptiPrep™ density gradient. We separated the five fractions from top to bottom. After ultracentrifugation, five individual fractions became apparent at the individual interphases and were collected, pelleted and used for subsequent analyses.
B   The Western blots were analyzed for the presence of known organelle marker proteins as indicated. Lysosomal and mitochondrial fractions from WT MEFs were enriched in fractions 2 and 5, respectively.
C   Lysate of WT MEFs was immunoprecipitated with anti-GAPDH, anti-PGK1, and IgG antibodies. Immunoprecipitants were analyzed by Western blotting with anti-PGK1 and anti-GAPDH antibody.
D   After adding various concentrations of NaCl to the lysosomal fraction of WT MEFs, the membrane pellet (bound) and supernatant (free) were subjected to immunoblotting with GAPDH and PGK1 antibodies.
E   Schematic representation showing that the lysosomal V-ATPase activity is regulated by the glycolytic enzymes, GAPDH and PGK1. The glycolytic GAPDH activity was measured with GAP as a substrate in the presence of NAD. PGK1 catalyzes the reaction of 1,3-bisphosphoglycerate (1,3-BPG) and ADP to form 3-phosphoglycerate (3-PG) and ATP.
F   After addition of GAP, NAD, ADP, and Pi, ATP production in the lysosomal fraction was measured using a luciferin/luciferase kit. Iodoacetate (IA; 4 μM) was used to inhibit GAPDH activity. Three independent experiments were performed. Error bars are presented as mean ± SEM of three independent experiments. Statistical significance was assessed by one-way ANOVA, ***$P < 0.001$.
G   Lysates were prepared from MEF cells expressing Tmem192-3 × HA [the only enzyme that catalyzes PI(3,5)P2 production in the lysosomal membrane] that were treated with 1 μM YM201636 (PIKfyve inhibitor). The lysosome immunoprecipitation method isolates pure lysosomes with anti-HA antibody. Immunoblotting of protein markers of various subcellular compartments in purified immunoprecipitated lysosomes are shown.
H   The lysosomal fraction of WT MEF cells using the Dextran magnetic beads (MAG10) method contained GAPDH and PGK1.
I    After addition of GAP, NAD, ADP, and Pi, ATP production in the lysosomal fraction isolated by MAG10 was measured using a luciferin/luciferase kit. Iodoacetate (IA; 4 μM) was used to inhibit GAPDH activity. Three independent experiments were performed. Error bars are presented as mean ± SEM of three independent experiments. Statistical significance was assessed by one-way ANOVA, **$P < 0.002$.
J    Schema of this novel study. Mitochondrial translation deficiency suppresses HIF1α-Nmnat3-mediated NAD$^+$ production. The glycolytic enzymes, GAPDH and PGK1, are associated with lysosomal vesicles, and NAD$^+$ is required for ATP production around lysosomal vesicles. NAD$^+$ content is essential for lysosomal maintenance.

Source data are available online for this figure.

however, HK2 and Hsp70 were absent (Fig 6G). Next, we investigated whether inhibition of PIKfyve (YM201636), the only enzyme that catalyzes PI(3,5)P2 production in the lysosomal membrane (Bissig *et al*, 2017), affects the localization of the glycolytic enzymes, GAPDH/PGK1. The addition of inhibitors did not significantly change the amount of glycolytic enzymes localized to lysosomes (Fig 6G), suggesting that PI(3,5)P2 does not affect the transport of glycolytic enzymes to lysosomes.

Second, after incorporation of Dextran magnetic beads (MAG10) (Lee *et al*, 2015), which accumulate in lysosomes, we purified the active organelles with a magnet and examined ATP production. Purification by magnetic beads showed that the lysosomal marker, Lamp2, was enriched and that GAPDH and PGK1 were present in this fraction, but HK2 was absent (Fig 6H). We also observed that ATP was produced in a GAP-dependent manner and was inhibited by IA (Fig 6I). To confirm that the observed vesicular acidification is exogenous ATP-dependent, the lysosomal vesicles were incubated with exogenous ATP. The lysosomal pH decreased by the ATP addition and the decrease was reversed with Concanavalin A, which inhibits lysosomal ATPases (Fig EV5F). These results suggest that the ATP generated by lysosomal vesicle-bound GAPDH/PGK1 is effective in maintaining the acidification of lysosomes.

To investigate the importance of local ATP production surrounding lysosomes, we investigated the acidification and function of lysosomes in high-glucose medium. High glucose provides the cytoplasm with a sufficient amount of ATP. High-glucose medium did not affect lysosomal function, whereas FK866-dependent NAD depletion reduced lysosomal function (Fig EV5G), suggesting that ATP was not supplied by high glucose-dependent glycolytic ATP production.

Taken together, we demonstrated that lysosomal fractions contain the glycolytic enzymes, GAPDH and PGK1, and revealed

that NAD$^+$ is essential for ATP production around lysosomes to maintain lysosomal acidification.

# Discussion

In this study, we showed that mitochondrial translation dysfunction leads to lysosomal dysfunction through reduced *Nmnat3* expression in the heart, suggesting a novel mechanism by which mitochondrial translation dysfunction inhibits lysosomal acidification (Fig 6J). Mitochondrial dysfunction due to excessive reactive oxygen species (ROS) appears to be a driving force of aging and disease (Wallace *et al*, 2010). Thus, autophagy is an essential cytoprotective pathway and a potential anti-aging mechanism (Madeo *et al*, 2010), suggesting that autophagic function may rescue cardiomyocytes of mitochondrial translation-deficient mice (p32cKO). Furthermore, it has been suggested that lysosomal dysfunction is impaired by mitochondrial translation deficiency and/or aging owing to reduced NAD$^+$ contents. Cardiomyocytes rely on autophagy, a lysosome-mediated degradation pathway, to remove aggregated proteins and damaged mitochondria (Kuma *et al*, 2004); thus, it is considered that lysosomal dysfunction due to reduced NAD$^+$ content affects heart failure.

It was reported that the chaperone-like protein p32 is an important regulator of ULK1 stability by forming a complex with ULK1 (Jiao *et al*, 2015). P32 knockdown in HeLa cells significantly impaired the clearance of starvation-induced autophagy flux and damaged mitochondria caused by mitochondrial uncoupler. We also found that p32 knockout in heart showed increased phosphorylation of ULK1 Ser757 and p62 Ser403 and also found that autophagic abnormalities, such as p62 accumulation, LC3 around broken mitochondria; however, we did not identify which level the autophagy flux is impaired in p32 knockout heart. In future studies, we will

investigate which stage of autophagy flux is involved in p32-deficient heart.

NAD$^+$ levels decrease in many organisms with age, resulting in reduced Sirtuin activity and downstream activation of autophagy (Hsu *et al*, 2009; Verdin, 2015). Subsequently, decreased NAD$^+$ levels impair lysosomal function and further reduce autophagic flux in several cell types (Hsu *et al*, 2009; Baixauli *et al*, 2015). NAD$^+$ levels increased in animal hearts by NMN addition led to increased Sirt1 activity and protection against ischemia/reperfusion injury (Yamamoto *et al*, 2014). Thus, NMN treatment is an effective way to activate autophagy in cardiomyocytes and improve heart health and longevity in mouse models. In future studies, we will investigate whether NMN treatment improves the lifespan of mitochondrial translation-deficient heart.

A direct link between the hypoxia-responsive transcription factor, HIF-1α, and nematode senescence has previously been reported (Leiser & Kaeberlein, 2010). This study has shown that HIF-1α can promote or limit lifespan via mechanistically distinct pathways. HIF-1α is a molecular marker of cellular aging and metabolism and is increased during aging. During aging, mitochondrial function and NAD synthesis gene expression were decreased, resulting in decreased NAD content. In the current study, we revealed a novel mechanism of mitochondrial function decline and NAD decline in aging via HIF1α. The next question is why does the mitochondrial translational defect stabilize HIF1α in our mouse model? Thus, further investigation is required to elucidate this pathway in more detail.

Fernandez-Mosquera *et al* (2019) have reported that mitochondrial respiratory chain deficiency affects lysosomal hydrolysis, resulting in lysosomal dysfunction. They demonstrated that mitochondrial dysfunction deactivates AMPK T172 phosphorylation (AMPK, a key regulator of energy homeostasis) signaling in tissue samples. Finally, they showed that downregulation of the AMPK-PIKFYVE-PtdIns(3,5)P2-MCOLN1 pathway inhibited Ca$^{2+}$ accumulation of lysosome and impaired lysosomal function. In our mouse model, we have previously observed increased AMPKα T172 and 4E-BP1 phosphorylation and decreased S6K phosphorylation, suggesting that the mTOR pathway is inhibited in p32-KO heart (Yagi *et al*, 2012). These differences are due to differential mitochondrial dysfunction between the respiratory chain and mitochondrial translation. We also observed that mitochondrial translational inhibition induced HIF1α stability and several genes involved in glycolysis or NAD$^+$ synthesis genes and, finally, decreased lysosomal acidification (Fig 3E–I). These results suggest that mitochondrial–lysosomal interactions use several signaling pathways that are mechanistically distinct.

Mitochondrial and lysosomal functions are intricately related and have important roles in maintaining cellular homeostasis (Hutagalung & Novick, 2011; Burté *et al*, 2015; Plotegher & Duchen, 2017). There is dynamic formation of organelle membrane contact sites between lysosome and mitochondria, suggesting that lysosomal or autophagic function affect the mitochondrial maintenance. For example, mTORC1 which inhibit autophagy induced integrated stress response in neurons (Khan *et al*, 2017) and regulates mitochondrial activity (Norambuena *et al*, 2018). Additionally, disrupting lysosomal acidification is sufficient for decreasing mitochondrial respiration (Monteleon *et al*, 2018). These results suggest that lysosomal function is linked to mitochondrial function.

In this study, we demonstrated that mitochondrial translation dysfunction led to decreased lysosomal acidification by reducing NAD$^+$ levels. NAD$^+$ was essential for ATP production around lysosomes and the GAPDH/PGK1 complex was associated with lysosomes (Figures 3–6), suggesting that reduced NAD$^+$ due to mitochondrial translation deficiency is an important cause of heart failure. Moreover, overexpression of Nmnat3 improves metabolic health and is an attractive therapeutic target for aging-related metabolic disorders (Gulshan *et al*, 2018). We showed that WT MEFs were highly dependent on mitochondrial ATP production, and p32KO MEFs were dependent on glycolytic ATP production (Yagi *et al*, 2012). p32KO MEFs showed less lysosomal activity and the total ATP levels were significantly higher in p32KO MEFs than in WT MEFs, suggesting that the difference in ATP production by mitochondria and glycolysis was not involved in lysosomal function. Therefore, local ATP production may be important for lysosomal function.

We found that *Nmnat* gene expression and activity decreased in response to impaired mitochondrial translation. Nmnat3, which is expressed in heart tissue, is localized in the mitochondria, but not in the cytosol (Stein & Imai, 2012; Verdin, 2015). A recent report has shown that Nmnat3 is localized in the cytosol and outside mitochondria (Yamamoto *et al*, 2016). We also found that in the heart, Nmnat3 is localized both in the cytosol and in mitochondria. Another study suggests the existence of an unrecognized mammalian NAD$^+$ (or NADH) transporter in mitochondria (Davila *et al*, 2018), though the mitochondrial inner membrane is considered impermeable to pyridine nucleotides, raising the possibility that Nmnat3 in the cytosol can also regulate the mitochondrial NAD$^+$ levels. Conversely, it has been suggested that cytosolic NAD$^+$ maintains the mitochondrial NAD$^+$ pool via cytosolic Nmnat3 (Felici *et al*, 2013).

Consistent energy is required to transport long neurons at high speed. As a result, it has been reported that it continues to fuel the molecular motors that transport vesicles. This group demonstrated that glycolysis provides ATP for the fast axonal transport of vesicles and vesicular GAPDH is necessary and sufficient for providing onboard energy for fast vesicular transport (Zala *et al*, 2013). GAPDH is localized to vesicles via a huntingtin-dependent mechanism and is transported by fast-moving vesicles in axons. These specially localized glycolytic mechanism can provide constant energy for the continuous movement of vesicles over long distances within axons (Zala *et al*, 2013).

Thus, we hypothesized that local ATP production by glycolytic enzymes near the lysosome is necessary for lysosomal function. The next question was whether ATP production by GAPDH depends on glucose uptake or NAD$^+$, because GAPDH uses GAP and NAD$^+$ as substrates. We demonstrated that high glucose did not affect the lysosomal acidification and protease activity, whereas NMN administration or Nmnat3 overexpression rescued lysosomal acidification, suggesting that NAD around lysosomes is required for local ATP production.

In this study, we observed that impaired mitochondrial translation induced HIF1α stabilization, Nmnat3 suppression and NAD$^+$ reduction, leading to lysosomal dysfunction. Thus, we propose a new molecular mechanism that links mitochondrial translational defects to lysosomal dysfunction via decreased NAD$^+$ levels.

# Materials and Methods

## Reagents and Tools table

| Reagent/Resource | Reference or Source | Identifier or Catalog Number |
| --- | --- | --- |
| **Experimental models** | | |
| p32cKO mice (αMHC-cre) | Saito *et al* (2017) | |
| p32KO MEF cells | Yagi *et al* (2012) | |
| **Recombinant DNA** | | |
| Nmnat2 | OriGene | MR204325 |
| Nmnat3 full | GeneScript | OMu17583D |
| Nmnat3 transcript varient1 | OriGene | RC206772 |
| TMEM192 | Addgene | 104434 |
| **Antibodies** | | |
| LC3A/B (D3U4C) XP® Rabbit mAb | Cell Signaling | 12741 |
| anti-p62(SQSTM1) pAb | MBL | PM045 |
| LAMP2 antibody [GL2A7] | abcm | ab13524 |
| GAPDH (14C10) Rabbit mAb | Cell Signaling | 2118 |
| anti-Phospho-p62(SQSTM1)(Ser351) pAb | MBL | PM074 |
| anti-Phospho-p62(SQSTM1)(Ser403) mAb | MBL | D343-3 |
| Phospho-eIF2α (Ser51) (D9G8) XP® Rabbit mAb | Cell Signaling | 3398 |
| Phospho-ULK1 (Ser555) (D1H4) Rabbit | Cell Signaling | 5869 |
| Phospho-ULK1 (Ser757) (D7O6U) Rabbit mAb | Cell Signaling | 14202 |
| ULK1 (D8H5) Rabbit mAb | Cell Signaling | 8054 |
| NMNAT-3 (D-10) antibody | Santa Cruz | sc-390433 |
| MTCO1 antibody [1D6E1A8] | abcm | ab14705 |
| Monoclonal Anti-HA antibody produced in mouse | Sigma-Aldrich | H3663 |
| Anti-HA-tag pAb-Alexa Fluor 488 | MBL | 561-A48 |
| DYKDDDDK Tag Monoclonal Antibody (L5), Alexa Fluor 488 | ThermoFisher | MA1-142-A488 |
| ANTI-FLAG® antibody produced in rabbit | Sigma-Aldrich | F7425 |
| PBEF/Visfatin/NAMPT Antibody | NOVUS | NB100-594 |
| Hexokinase II (C64G5) Rabbit mAb | Cell Signaling | 2867 |
| Anti-PGK1 antibody [EPR19057] | abcm | ab199438 |
| V-ATPase B1/2 Antibody (F-6) | Santa Cruz | sc-55544 |
| Anti-Multi Ubiquitin mAb | MBL | D058-3 |
| Anti-Pyruvate dehydrogenaseE2/E3bp antibody | abcm | ab110333 |
| Anti-HIF-1 alpha antibody | abcm | ab179483 |
| Anti-Lactate Dehydrogenase antinody | abcm | ab47010 |
| Anti-NMNAT2 antibody | abcm | ab56980 |
| DAPI solution | Dojindo | D532 |
| p32 | Yagi *et al* (2012) | |
| VDAC | Yagi *et al* (2012) | |
| **Oligonucleotides and sequence-based reagents** | | |
| PCR primers | This study | Table EV1 |
| **Chemicals, enzymes and other reagents** | | |
| HIF1α inhibitor | MERCK | 934593-90-5 |
| DexoMAG40 | Liquid Research Ltd | F2288 |

**Reagents and Tools table** (continued)

| Reagent/Resource | Reference or Source | Identifier or Catalog Number |
|---|---|---|
| GAP (Glyceraldehyde 3-phosphate) | SIGMA | G5251 |
| ADP | SIGMA | A2754 |
| NAD | SIGMA | N0632 |
| FK866 | Selleckchem.com | S2799 |
| Bafilimycin A | BioViotica | NSC381866 |
| LysoSensor™ Yellow/Blue DND-160 | Invitrogen | L7545 |
| IA (Sodium iodoacetate) | SIGMA | I2512 |
| DQ-BSA | Thermo Fisher Scientific | D12050 |
| Dextran TMR | Invitrogen | D1868 |
| Dextran Green | Invitrogen | D7170 |
| ACMA (9-amino-6-chloro-2-methoxyacridine) | Invitrogen | A1324 |
| YM201636 | Cayman | 13576 |
| Concanamycin A | BioViotica | BVT-0237 |
| DAPRed | DOJINDO | D675 |
| DALGreen | DOJINDO | D676 |

## Methods and Protocols

### Animals

Animal care was in compliance with the Kyushu University animal care guidelines (#A26-044). All experimental procedures conformed to the Guide for the Care and Use of Laboratory Animals, Eighth Edition, updated by the US National Research Council Committee in 2011. The animals were treated in accordance with the guidelines stipulated by the Kyushu University Animal Care and Use Committee (Saito *et al*, 2017). Mice homozygous for exon 3 floxed *p32* allele (*p32$^{flox/flox}$*) were generated as described previously (Yagi *et al*, 2012). To obtain heart-specific p32cKO mice, we crossed *p32$^{flox/flox}$* mice with αMHC-cre (Myh6 promoter) mice (Saito *et al*, 2017).

### Cell lines and reagents

MEFs (WT and p32KO) were obtained in our laboratory (Yagi *et al*, 2012). All cell lines were cultured in Dulbecco's modified Eagle medium (DMEM; 1,000 mg/l glucose; Sigma–Aldrich, St. Louis, MO, USA) supplemented with 10% FBS at 37°C in a humidified atmosphere with 5% $CO_2$. MEFs were incubated with the Nampt inhibitor, FK866 (10 nM; Selleck Chemicals Llc, Houston, TX, USA), for 24–48 h in 5% $CO_2$ at 37°C, followed by measurement of total cellular $NAD^+$ and NADH content using the NAD/NADH-Glo™ Assay (Promega). YM201636 (13576) was purchased from Cayman Chemical. These compounds were solubilized in DMSO (Sigma–Aldrich, D8418), which was used as vehicle control.

### Immunoblotting

Heart tissues were immediately frozen in liquid nitrogen. Tissues and cultured cells were lysed with lysis buffer [20 mM Tris–HCl, pH 7.5, 150 mM NaCl, 2 mM EDTA, 1% NP-40, 0.1% SDS, protease inhibitor cocktail (WAKO)], homogenized by sonication and then subjected to immunoblotting as described previously (Yagi *et al*, 2012).

### Immunohistochemistry of heart sections

After mice were anesthetized with an overdose of sevoflurane, tissue sections were prepared from hearts fixed in 10% formaldehyde to obtain paraffin-embedded coronal sections, which were stained with various antibodies. Argon laser light (488 nm, 540 nm) was used to excite lipofuscin autofluorescence. A band-pass filter between 510–560 nm or 580–630 nm was used for autofluorescence. To reduce the lipofuscin-like autofluorescence, we used TrueBlack™ (Biotium) reagents before primary antibody treatment.

### Electron microscopy

For electron microscopy, samples were fixed with 2% paraformaldehyde (PFA) and 2% glutaraldehyde (GA) in 0.1 M phosphate buffer (PB), pH 7.4 at 4°C overnight. The method details have previously been reported (Saito *et al*, 2017).

### Real-time PCR analysis

Total RNA was extracted using the ReliaPrep™ RNA Tissue Miniprep System (Promega). cDNA was synthesized using total RNA, random hexamer primers, oligo dT primers, and the PrimScript™ RT reagent Kit (TaKaRa). The cDNA was then subjected to real-time PCR analysis with SYBR Premix Ex Taq™ II (TaKaRa) and the StepOnePlus Real-Time PCR System (Applied Biosystems). The expression level of each mRNA was normalized to the level of 18S ribosomal RNA obtained from the corresponding reverse transcription product. Primer sequence was listed in Table 1.

### Metabolome assay

Heart-derived metabolites were analyzed by LC-MS/MS based on reverse phase ion-pair chromatography and hydrophilic interaction chromatography modes coupled with the triple quadrupole mass spectrometer, LCMS-8040 (Shimadzu, Kyoto, Japan) as described previously (Saito *et al*, 2017).

### Nmnat activity

Mouse hearts were homogenized in a BioMasher tube (Nippi) containing 50 mM HEPES-KOH, pH 7.4, 0.5 μM EDTA, 1 mM MgCl$_2$, and protease inhibitor cocktail; samples were then centrifuged at 800 $g$ for 10 min at 4°C. The supernatant was measured for Nmnat activity as described previously (Conforti *et al*, 2007). Briefly, 5 μg of supernatant protein in 50 μl of reaction buffer (30 mM Tris–HCl, pH 7.5, 2 mM ATP, 20 mM MgCl$_2$) were added to 2 mM NMN to start the activity assay. After incubation at 37°C for 15 min, NAD was quantified using the NAD/NADH-Glo™ Assay (Promega).

### Lysosomal content assay

MEFs were stained with Dextran, Oregon Green™ 488, 10,000 MW, Anionic (quenched under acidic environment) and Dextran, Tetramethylrhodamine, 10,000 MW, Anionic, Lysine Fixable (pH independent) (Thermo Fisher) to measure lysosomal acidification. After addition of the two dextran molecules, cells were incubated for 6 h at 37°C in a glass bottom dish (IWAKI) and incubated overnight after medium change. The medium of the labeled cells was replaced with HBSS, and the cells were observed under a fluorescence microscope. The intensity of green and red fluorescence was measured and calculated (BZ-X800, KEYENCE).

In another approach, we used LysoSensor™ Yellow/Blue DND-160 (PDMPO) (Molecular Probes) to measure lysosomal pH. MEFs were seeded in glass bottom culture dishes (IWAKI) and incubated in 25 μM monensin for 20 min, and then 1 μM LysoSensor was added for 5 min, followed by two washes with PBS. Images were captured by confocal microscopy (TCS SP8, Leica). To calibrate the standard curve for LysoSensor pH, MEFs were incubated in buffer with different pH values (5 mM NaCl, 115 mM KCl, 1.2 mM MgSO$_4$, 25 mM MES, pH 3.5–7.5). Cells were excited with a range of UV wavelengths (357–373 nm) and observed in the blue (W1; 417–483 nm) and yellow (W2; 490–530 nm) regions of the spectra. The ratio (W1/W2) images were generated using the image calculator function in MetaMorph (MOLECULAR DEVICES).

### Rescue by Nmnat2 and Nmnat3 plasmid transfection

The *Nmnat2* expression plasmid was purchased from GeneScript. The plasmids harboring *Nmnat3* with or without MTS were purchased from OriGene. These plasmids were transfected into p32KO MEFs by Lipofectamine™ LTX reagent (Thermo Fisher). For FACS analysis, transfected MEFs were incubated with 500 nM LysoTracker Red DND-99 (Molecular Probes) for 1 h in 5% CO$_2$ at 37°C, fixed with 4% Paraformaldehyde Phosphate Buffer Solution (nacalai) for 10 min and then permeabilized with 0.2% Triton X-100/PBS for 5 min. Samples were blocked with Blocking One (Nacalai) and then incubated with a 1:500 dilution of anti-HA-Alexa Fluor 488 (MBL) or anti-Flag-Alexa Fluor 488 (Thermo Fisher) in Blocking One (Nacalai) for 1 h. Flow cytometry analysis was performed with a FACSVerse using FACSuite software (BD Biosciences). For imaging, glass slides were mounted using Mowiol (Sigma-Aldrich). Fluorescence images were obtained using a fluorescence microscope (BZ-9000, KEYENCE).

### Immunohistochemistry

MEFs were incubated with 10 nM FK866 (Selleckchem) for 24–48 h in 5% CO$_2$ at 37°C. Immunofluorescence was carried out according to established techniques (Mikako Yagi 2012). MEFs were fixed with 4% Paraformaldehyde Phosphate Buffer Solution (Nacalai) for 10 min and permeabilized in 0.2% Triton X-100/PBS for 5 min. Samples were blocked with Blocking One (Nacalai), and the primary antibody was diluted with Can Get Signal® (TOYOBO).

### DQ-BSA

DQ™ Green BSA (Thermo Fisher) was used for lysosomal protease activity. A total of $1 \times 10^4$ cells/well were cultured overnight in a 96-well plate. DQ™ Green BSA (50 μg/ml) was added and incubated for 1 h at 37°C, and then, the culture medium was replaced with PBS and the fluorescence at 495 nm excitation and 525 nm emission was measured by ARVO (PerkinElmer).

### Immunoprecipitation

MEFs were incubated with 1% formalin/DMEM for 10 min in 5% CO$_2$ at 37°C. Glycine (1 M) was added to the harvested cells to quench cross-linking. The cell lysate was suspended in PBS with 1% Triton X-100, sonicated, and centrifuged at 20,000 $g$ for 10 min. The lysate was incubated with anti-PGK1 (Abcam) or anti-GAPDH (Cell Signaling Technology) antibodies. As the negative control, we used IgG (Cell Signaling Technology). After 16 h of rotation, protein A/G was added and incubated for 2 h, and the lysate was washed four times with PBS and eluted with SDS–PAGE sample buffer. The elution was boiled for 30 min to reverse cross-linking.

### Lysosomal fraction

MEFs were harvested by trypsin, homogenized in homogenization buffer (250 mM sucrose, 20 mM HEPES-KOH, pH 7.4, 1 mM EDTA, protease inhibitor cocktail), and centrifuged at 800 $g$ for 10 min to remove nuclei. The supernatant was centrifuged at 20,000 $g$ for 30 min, and the pellet (membrane fraction) was resuspended in 19% ($v/v$) OptiPrep™ solution (AXS: Axis-Shield Density Gradient

**Table 1.** List of primer used in this study.

| Target | Forward (5′->3′) | Reverse (5′->3′) |
|---|---|---|
| LC3 | cgtcctggacaagaccaagt | attgctgtcccgaatgtctc |
| Gabarap1 | catcgtggagaaggctccta | atacagctggcccatggtag |
| mvps34 | tgtcagatgaggaggctgtg | ccaggcacgacgtaacttct |
| atrogin1 | tgggtgtatcggatggagac | tcagcctctgcatgatgttc |
| Lamp2a | tggctaatggctcagctttc | atgggcacaaggaagttgtc |
| Beclin | ggccaataagatgggtctga | cactgcctccagtgtcttca |
| Atg4b | attgctgtggggttttttctg | aaccccaggatttttcagagg |
| Atg12l | ggcctcggaacagttgttta | cagcaccgaaatgtctctga |
| Ulk2 | cagccctggatgagatgtttt | ggatgggtgacagaaccaag |
| Nmnat1 | gaagtgggctgatcaaaagc | ccagcccgagtgatacagat |
| Nmnat2 | ctctggctcttgggtttctg | acagctctggagatggccta |
| Nmnat3 | tccagcagtttcagcacaac | gaggccctctagccagtctt |
| nampt | tacagtggccacaaattcca | caattcccgccacagtatct |
| Naprt | tgccctggctagagtctgtt | tcagggtcctctgtcagctt |
| Qprt | agtcaccatggaccctgaag | gaagatggcgtcaaagaagg |
| Sirt3 | tcctcgaaggaaagatgtgg | gcatgaagtcttgctggaca |
| Nadsym1 | accggaatgttcgctacaac | catctccaaagggcacagtt |
| 18S | cgcggttctattttgttggt | agtcggcatcgtttatggtc |

Media—a brand of Alere Technologies AS) as described previously (Schmidt *et al*, 2009). The sample was loaded on a discontinuous density gradient with 27%, 22.5%, 19%, 16%, 12%, and 8% Opti-Prep™ (2 ml each) in a 12-ml centrifugation tube and subjected to ultracentrifugation at 130,000 $g$ for 4 h at 4°C in a swinging bucket rotor (SW 40.1; Beckman Coulter). The subcellular fractions were collected from the top of the tube, washed and concentrated with homogenization buffer, and centrifuged at 20,000 $g$ for 30 min. The protein content of the individual fractions was determined by Western blotting.

### GAPDH and PGK1 binding to lysosomes

For analysis of GAPDH and PGK1 binding to lysosomal membranes, purified lysosomal fractions (Fraction No.2) were divided into 5 parts and resuspended in homogenization buffer (250 mM sucrose, 20 mM HEPES-KOH, pH 7.4, 1 mM EDTA) containing each 0, 0.1, 0.2, 0.4, 0.8 M NaCl. The mixtures were incubated for 30 min on ice and subjected to ultracentrifugation at 160,000 $g$ for 1 h at 4°C in a fixed angle rotor (TLA120.2; Beckman Coulter). The pellet (lysosomal membrane) were resuspended in 50 μl of SDS–PAGE sample buffer and resolved by SDS–PAGE. The supernatant (protein off lysosomal membrane) were added 20% Trichloroacetic acid (TCA) to extract protein for 30 min and centrifuged at 20,000 $g$ for 10 min at 4°C. The pellet (extract protein) was washed with cold acetone, centrifuged at 20,000 $g$ for 5 min at 4°C. The TCA/acetone precipitate was also suspended in 50 μl of SDS–PAGE sample buffer and resolved by SDS–PAGE.

### GAPDH and PGK1-dependent ATP production

Five micrograms of the lysosome fraction were resuspended in 200 μl of reaction buffer [20 mM HEPES-KOH, pH 7.2, 200 mM sucrose, 50 mM KCl, 1 mM NAD, 0.5 mM ADP, 1 mM $KH_2PO_4$]. To start measuring ATP production, 1 mM glyceraldehyde-3-phosphate (G3P) and 10 μM Iodacetate (Sigma-Aldrich), which is a GAPDH inhibitor, were added. The reaction was incubated at room temperature for 1 min, and then, 100 μl of the reaction mixture and 100 μl of CellTiter-Glo reagent were mixed in a 96-well plate. After 10 min, luminescence was recorded [CellTiter-Glo Luminescent Cell Viability Assay (Promega)] (Zala *et al*, 2013).

### Lysosome isolation by MAG10

3T3-L1 cells were incubated with 0.01% DexsoMAG 10 (Liquid Research Limited)/D-PBS(−) (Nacalai) in 5% $CO_2$ at 37°C for 16 h and then incubated in DMEM containing 10% FBS for 24 h. Cells were harvested by trypsinization, homogenized in 1 ml of buffer A (1 mM HEPES-KOH, pH 7.2, 15 mM KCl, 1.5 mM MgAc, 1 mM DTT, protease inhibitor cocktail) using a Dounce homogenizer (25 strokes), and passed through a 27-G needle five times. A total of 125 μl of buffer B (220 mM HEPES-KOH, pH 7.2, 375 mM KCl, 22.5 mM MgAc, 1 mM DTT, protease inhibitor cocktail) was added, followed by centrifugation at 800 $g$ for 10 min to remove nuclei. The supernatant was loaded on an LS column (Miltenyi) that was pre-equilibrated with 0.5% BSA/PBS. DNase I (QIAGEN) solution was added to the LS column for 10 min and then washed three times with 3 ml of PBS. The column was removed from the separator, and the magnetically labeled cells were flushed out immediately by firmly pushing the plunger; the elution contained enriched lysosomes (Lee *et al*, 2015).

### v-ATPase activity

Isolated lysosomes were incubated in reaction buffer [1 mM MOPS-Tris, pH 7.2, 100 mM KCl, 12.5 mM $MgCl_2$, 2 μM ACMA (9-amino-6-chloro-2 methoxyacridine) (Invitrogen)] at 37°C for 3 min. Then, 1 mM ATP was added, and the rate of initial fluorescence quenching was monitored with ALVO™ X2 (PerkinElmer). ConA (con-canamycin A; 1 μM; AdipoGen) was added to inhibit v-ATPase. The excitation/emission wavelengths were set to 355 and 460 nm, respectively.

### Fluorescence probes

DALGreen and DAPRed (Dojindo) stain autophagosomes with fluorescence enhancement in a hydrophobic environment. DALGreen's fluorescence is enhanced at acidic pH and is suitable for monitoring the autophagy degradation stage (autolysosome). In contrast, DAPRed has a pH-independent fluorescence profile and remains fluorescent with almost constant intensity throughout the process of autophagy (autophagosome). MEFs seeded in glass bottom dishes were stained with DALGreen and DAPRed for 30 min at 37°C. The medium was replaced with HBSS for 5 h to induce autophagy. Then, cells were cultured with 10 nM FK866. To inhibit lysosomal function, cells were treated with 50 μM bafilomycin A1. The intensity of the green and red signals was measured and calculated (BZ-X800, KEYENCE).

### Quantification and statistical analysis

The data are expressed as means ± SEM of the indicated number of experiment and mice. Unpaired Student's $t$-test was used to determine statistical differences between two groups. *$P < 0.05$, **$P < 0.01$, or ***$P < 0.005$ was considered statistically significant.

# Data availability

This study includes no data deposited in external repositories.

**Expanded View** for this article is available online.

## Acknowledgements

This work was supported by the JSPS (17H01550, 15H04764, 25253041, 18K15421). We thank Edanz Group (https://en-author-services.edanzgroup.com/ac) for editing a draft of this manuscript.

## Author contributions

Research design and data analysis: MY, TU; Experiments and data analysis: MY, TU, TT, YD, HH, RA, DS; Manuscript writing: MY, TU, and DK.

## Conflict of interest

The authors declare that they have no conflict of interest.

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
