## [Review Process File · The EMBO Journal]

Mitochondrial translation deficiency impairs NAD⁺-mediated lysosomal acidification

Author information redacted

DOI: [10.15252/embj.2020106663](https://doi.org/10.15252/embj.2020106663)

Corresponding author(s): Author information redacted Author information redacted (Author information redacted) Author information redacted

Review Timeline:

Submission Date:	13th Apr 20
Editorial Decision:	6th May 20
Revision Received:	29th Aug 20
Editorial Decision:	1st Oct 20
Revision Received:	18th Oct 20
Editorial Decision:	10th Nov 20
Revision Received:	21st Nov 20
Accepted:	8th Dec 20

Editor: Elisabetta Argenzio

Transaction Report:

Thank you for submitting your manuscript entitled "Mitochondrial translation deficiency impair NAD mediated lysosomal acidification and autophagy" [EMBOJ-2020-105268] to The EMBO Journal. The study has now been seen by three referees, whose comments are provided below. In light of these reports, I am afraid that the manuscript is not a sufficiently strong candidate for publication here.

As you can see, while the referees find the study potentially interesting, they are concerned that the main conclusions are not supported by the experimental data and that the methodology used to obtain lysosomal fractions is not adequate for this particular study.

Given these opinions from trusted experts in the field and the large amount of new experiments required to substantiate the proposed model of impaired mitochondrial translation preventing lysosomal acidification, we are regretfully unable to invite a revised version of your manuscript.

However, considering the potential interest of your findings, I would be willing to reconsider your study as a new submission at a later time if the referees' concerns would be fully addressed and their suggestions implemented. Please note that the novelty of the manuscript will be assessed at the time of re-submission.

While I am sorry that I cannot communicate more positive news for The EMBO Journal, I hope that

Referee #1:

The manuscript "Mitochondrial translation deficiency impair NAD mediated lysosomal acidification and autophagy" proposes that a genetic defect in mitochondria (KO of the protein p32, which causes defects in mitochondrial translation) results in decreased cytoplasmic NAD⁺, which in turn results in decreased production of ATP at the lysosome, with the ultimate consequence of lower activity of the lysosomal vATPase and therefore impaired lysosomal acidification.

The concept is very interesting, and centered on the nascent field of mitochondria-lysosome crosstalk. However, I have several concerns related to novelty and methodology.

First and foremost, the authors simply chose to completely ignore the published studies on how mitochondrial defects impair lysosomal function. This is a severe flaw of the current manuscript, and I urge the authors to correct it - including to support their own findings. Specifically, mitochondrial defects of different nature (TFAM, respiratory chain, mitochondrial dynamics) have been shown to

impair lysosomal biogenesis and function (PMID: 26299452, 26987902, 30917721, 28345620; a recent review summarizes other studies: 31791731). While the authors are correct that defects specific to mitochondrial translation have not yet been shown to impair lysosomes, this is hardly unexpected given the different models of mitochondrial malfunction that were already shown to impair lysosomes. Secondly, the study by Baixauli et al (26299452), which explores how mitochondrial malfunction caused by absence of TFAM results in lysosomal deficiency, already proposed that supplementation of NAD⁺ in that context can rescue some aspects of lysosomal function. Therefore, the novelty of the current paper is limited to the mechanism by which NAD⁺ can rescue lysosomal function. And, should the authors refer the previous studies and correct the methodological details, I consider that this is enough novelty to warrant publication.

My methodological concerns are detailed below, but they can be summarized in few words: the method that the authors used to obtain lysosomal fractions is not adequate for this particular study because it is not possible to exclude the possibility that there are cytoplasmic contaminants. This is a fundamental flaw, given that the model proposed here posits that it's two cytoplasmic proteins that attach to the lysosome that are involved in the synthesis of the ATP that is needed for vATPase function. The last experiment is made with magnetic-purified lysosomes - the others need to be as well.

Finally, two experiments are, in my view, missing to prove the model that the authors propose. If the problem of the vATPase is that there is not enough ATP, why can't it take ATP from the cytoplasm? Does this also happen in high-glucose medium, i.e., is it that there is not enough ATP production in this low glucose-medium? This is a fairly simple experiment and would be informative. The second has to do with the Nmat3 that the authors propose locates to the lysosome (my detailed discussion about the problem with that experiment is below): what makes it go to the lysosome? A study has shown that lysosomal malfunction caused by mitochondria is due to lower PI(3,5)P₂ in the lysosome. Could this affect the recruitment of Nmat3, or of the glycolytic proteins, to the lysosome? The authors could easily test if inhibition of PIKfyve, the only enzyme that makes PI(3,5)P₂ in the lysosomal membrane, would affect the localization of the glycolytic enzymes or of Nmat3 (or the expression of Nmat3).

"When cells are deprived of nutrients, to maintain metabolic homeostasis and cell viability, mitochondria can also be degraded non-selectively together with other cytosolic contents (Youle & Narendra, 2011)." - In fact, mitochondria are known to be protected from autophagy during starvation, via hyperfusion (PMID 21478857, PMID 21646527).

"no clear relationship between mitochondrial dysfunction and lysosomal function has been revealed." - this is simply not true. The interdependence between mitochondria and lysosomes, and particularly how dysfunctional mitochondria perturb lysosomal function has been studied by several labs (PMID 26299452, 26987902, 30917721, 28345620, 31791731), and I don't see how the authors chose not to incorporate this literature into the manuscript.

"The expressions of autophagy-related Gabarapl1, Atrogin1, Lamp2, and Atg4b genes were increased in the heart of p32cKO mice compared with that in WT mice (Figure 1G)." - the figure shows that Atrogin1 is down, the others are up. The text should be corrected accordingly.

Figure 2B - autofluorescence appearance? Number of autofluorescent particles or...?

"Collectively lysosomes and autolysosome in p32cKO hearts seem functionally impaired." - from figures 2A-2C, we can conclude

that the lysosomes show morphological signs of dysfunction. To conclude that they are functionally impaired more experiments are required, i.e., which aspect of lysosomal function is actually impaired.

Figure 2E - why VDAC (mitochondrial outer membrane) as a loading control for a whole cell extract? Especially given that the p32-KO cells have a mitochondrial defect...

Figure 2G - the representative western for ULK1-555P does not match the quantification

"The phosphorylation of ULK1 Ser757 (moles per mole) from p32cKO heart was increased 4-fold compared with WT samples, in agreement with the overall increase in mTOR signaling (Figure 2G)." - there is no assessment of mTORC1 activity (the mTOR complex that regulates ULK1 is the mTORC1, so that rather than mTOR should be mentioned. Furthermore, in the study that shows the effect of p32 on mitochondria (Saito et al, Cardiovascular Research, 2017, 113: 1173-1185), the authors conclude that mTORC1 is inhibited. Therefore, it is important that the authors clarify what happens to mTORC1, and if mTORC1 is down, why would ULK1 be more phosphorylated in the site targeted by mTORC1 (Ser757).

"Taken together, we consider that the degradation mechanism by autophagy was rather more strongly initiated but stopped midway" - this conclusion is at odds with the data. The ULK1 phosphorylation at Ser757 is inhibitory of the formation of autophagosomes at a very early stage. So it cannot be concluded that autophagy was strongly initiated... The "stopped midway" likely refers to a stalling of the autophagic flux, which is likely caused by the lysosomal impairment. This should be rewritten to fit what the data shows.

"To know which Nmnat isozyme is responsible, we examined the expression and localization of Nmnat2 and Nmnat3 in cardiac tissue by western blotting." - the authors should clarify why Nmnat1 was not examined.

Figure 3D - the method used is rather unclean. There are much cleaner and accurate methods established that allow the dissection of lysosomal and mitochondrial proteins fairly easily. Given that this is an important point that the rest of the study builds on, I urge the authors to prepare clean mitochondrial and lysosomal preps (for example, via immunoisolation, see PMID), and show that Nmnat3 is present in the organelle preps. As such, the method does not allow for an unequivocal interpretation. A similar concern regarding Figure S1B, with the added question that if the authors want to build on cytoplasmic localization of Nmnat3, then the western that they show also shows leakage of VDAC, which is an outer mitochondrial membrane protein. This suggests that there was contamination of the cytoplasmic prep with mitochondria, and begs for the levels of some soluble mitochondrial protein to determine the level of leakage. (also, important to distinguish cytoplasmic soluble versus lysosomal)

Figure 3F - there are no error bars or statistical treatment, not possible to make a conclusion on the result

"To examine lysosomal acidification, we used LysoTracker, a qualitative acidic pH indicator dye that is enriched and strongly fluoresces in compartments with a pH below 5 (Lee, Yu et al., 2010). LysoTracker staining was weak in p32-knockout MEFs compared with WT cells, suggesting that lysosomal acidification was reduced in p32-knockout MEFs (Figure 4A)." - as the authors correctly point out, LysoTracker is only a qualitative indicator. The biggest issue here is that there is no control for the amount of LysoTracker that is loaded to the lysosome. So the signal observed may be a reflection of higher pH or of lower uptake by the lysosome, or both. A more quantitative

method should be employed, such as the green/red dextran approach (two molecules of the same size, to control for delivery, and two different fluorophores, one pH-sensitive and the other - insensitive; the ratio reflects solely the luminal pH). Furthermore, the authors fixed the cells after lysotracker treatment, which is not indicated as it affects the results. Finally, I cannot find in the methods or figure legend how long was NMT added before imaging - it should be added (and how much, and what was the vehicle). This is also a concern for the other panels of Figure 4 that rely on lysotracker, as well as Fig S3.

Figure 4B - the overexpression in the WT cells should also be shown, inclusive on the pH experiments, so that it can be assessed if the rescue of the KO brings it to WT levels or somewhere in between - and overexpression of NMNAT affects the WT.

Figure 5 - the images have too much staining background to allow for proper colocalization quantification. It is important that the authors show the individual channels as well, so that one can assess the LAMP2 staining (very clear on the control, but impossible to assess on the others). Once the individual channels are presented, it will be possible to have a better overview of this result.

"After bafilomycin A treatment, an increase in DALGreen staining as well as fusion of DAPRed and DALGreen staining was observed, suggesting
16

a reduction in autophagic degradation within autolysosomes (Figure 5D)." - this is a puzzling observation. The treatment with bafilomycin is expected to neutralize the lysosomal pH, so why would DALGreen, which fluoresces more in acidid pH, be enhanced? As mentioned before, the duration, concentration and vehicles of the treatments have to be indicated. It is hard to assess this experiment from a steady-state point-of-view. If the effect of the FK866 is fast (and given that NAD⁺ is synthesized and used recurrently, it is expectable that the effect is fast), it would be much more informative to show this experiment using live imaging, and present the videos or a time-course. (I read now figure S5, in which the timeframe for FK866 treatment is indicated as 24-48h. This is quite a long treatment. How long after the FK866 is added to the cells do the concentrations of NAD⁺ start decreasing?)

"These results indicate again that inhibition of NAD⁺ synthesis impairs lysosomal function and resultantly suppresses degradation by autolysosome" - to make statements on lysosomal degradation, it is necessary to assess it in more detail, by monitoring for example cathepsin activities with Magic Red.

Figure 6 - cleaner method to isolate lysosomes needed, as mentioned above.

"p62 was also found in the lysosomal fraction, suggesting that fractions 2 and 3 represent the lysosome and autolysosome fractions" - or other structures, there is no way to be sure that this is only lysosomes. That is why it is pivotal to make a pure lysosomal prep.

"The analysis of these subcellular fractions indicated that GAPDH and PGK are in the cytosol and can bind to various types of membranes such as lysosomal membranes." - or that there are cytoplasmic contaminants in all fractions.

"We next evaluated whether GAPDH and PGK are in a complex in the cell by co-immunoprecipitation experiments. HeLa cells transfected with a GAPDH-HA plasmid were lysed" - why the change from MEFs to HeLa cells for this particular experiment? MEFs were transfected before in the paper, so transfection shouldn't be an issue. Furthermore, the fact

that the IP is performed using overexpression of one of the partners, begs for the reverse IP. Both should be done in MEFs or cardiomyocytes, to ensure that the system is operative in the same cells where the phenotype was described.

"We incubated the lysosomal vesicles fraction with GAP, ADP, Pi, and NAD to measure ATP production. We observed significant GAP-dependent ATP production under these conditions (Figure 6E), showing that both GAPDH and the downstream PGK are enzymatically active in the lysosomal fraction" - this goes back to the previous point in which it was not possible to assess if the presence of GAPDH and PGK was due to cytoplasmic contamination of fractions containing lysosomal components. The preparation of pure lysosomes is fundamental for the conclusions. For example, using the magnetic bead strategy employed in 6F.

"These results suggest that mitochondrial dysfunction due to mitochondrial translation deficiency is not directly involved in autophagy and severe heart failure. Both of the ATG5cKO mice and ATG5/p32 dcKO mice died at seven months, suggesting that autophagy failure was a major cause of p32cKO mice death (Figure 7H)." - It has been shown that lysosomal malfunction triggered by defects in mitochondria is not due to autophagosome delivery to the lysosomes (PMID). This experiment should therefore be placed in the context of published literature.

"Finally, these mice suggest that autophagy dysfunction due to mitochondrial dysfunction in hearts leads to severe heart failure and death." - this sentence is at odds with the previous sentence, and should be corrected.

The DRP1 chapter on the discussion is out of place and doesn't really make sense with the rest of the manuscript.

Once more, these findings should be placed in the context of the published literature.

Referee #2:

In the manuscript "Mitochondria translation deficiency impairs NAD⁺ mediated lysosomal acidification and autophagy" the author describes the physiological role of p32 in cardiomyocytes. The manuscript is composed of several parts. At the beginning the authors described the role of p32 in heart. The absence of p32 determines a phenotype with the presence of abnormal mitochondria and lysosome. These two features could be explained with a problem in the autophagy flux. A second part is dedicated to the role of NAD⁺ in p32 deficient cells and in particular the authors focus on Nmnat3 gene expression and positive function of the cytosolic form of this gene. They also investigate the beneficial effect of FK866 on lysosomes and autophagy. A third part is more on the molecular mechanisms and in particular regarding the ATP production on lysosomal surface. The last part looks back to the autophagy impairment. The manuscript has interesting points; therefore, the huge amount of information sometime is more confusing than clarifying. It is difficult to follow what the authors really want to say and what is the final message. Many scientific questions have been opened but only few of them have been really addressed in details. There are still points that need to be properly addressed in terms of experimental data and discussion. Moreover, the impairment of autophagy, that leads to mitochondria accumulation, is a well-known concept. The fact that p32 knockout negatively affects

autophagy flux and the mice hearts present abnormal mitochondria is quite awaited.

Major points:

1. Figure 1: the fact that mitochondria are surrounded by Ub, p62 and LC3 is not sufficient to state that they are inside autophagosomes. Electron microscopy images in the heart tissue or cytofluorimetry assay in cardiomyocytes using mKeima and/or RFP-GFP probes linked to mitochondria proteins would be more informative.

2. Page 8 paragraph "Lysosomal morphology is impaired in the heart of p32-knockout mice". This title is a bit misleading. Morphology could be altered or abnormal, function can be impaired. Both, lysosomal morphology and function, seem to be affected in p32-knockout, however, more experimental evidences are needed.

3. Figure 2: abnormal lysosomes are clear. What should be described more into details is the presence of mitochondria inside autolysosomes. Fig 2C is not really informative. Moreover, to claim that there is an autophagy flux problem, which likely is present, author should provide additional autophagy flux assays in order to exactly point where the problem is.

4. The NAD⁺ regulation and its role in lysosomal acidification is interesting. Data presented indicate that p32 has an effect on Nmnat3 and on lysosomes too. This part is interesting therefore few more experiments oriented to better elucidate the signaling pathway p32-HIF1 α -Nmnat3. The use of FK866 is of potential interest as a modulator of lysosomal function and autophagy flux. Therefore, more specific and detailed experiments are needed to support that lysosomal activity and autophagy flux have been restored. The reported data are not sufficient to make solid conclusions.

5. The paragraph regarding the molecular mechanisms linking GAPDH, PGK, NAD⁺ and ATP production at the lysosome surface is interesting and actually point to the role of the lysosome as signaling hub. This part deserves a more critical analysis in terms of experimental data. Conclusions are interesting but not fully supported by data.

6. Last paragraph pages 20-21 can be fused together with the first one. In both cases the authors are describing impaired autophagy phenotype.

7. Discussion is too long and lengthy. It is difficult to follow what the authors want to point out. I would suggest to significantly shorten the text and clearly focus on the main message of the manuscript.

The manuscript has potential but it is difficult to catch the message that the authors want to deliver. In order to be considered for publication the manuscript should be reorganized to be simpler to read and more focused to the final message. Several considerations regarding the role of autophagy in heart cells could be shortened and data moved to supplementary. Autophagy in heart has been investigated in the past. Mitochondria translation deficiency, that is the pointed in the title, should have a more relevant position in the manuscript and more experimental data in its support. The crosstalk between mitochondria and lysosome is also interesting and it could be highlighted and studied deeper.

Referee #3:

This study sets out to investigate how autophagy is impacted by defective mitochondrial translation using a p32 KO mouse model. The mouse has previously been used to show that loss of p32 causes a severe mitochondrial translation defect and respiratory chain deficiency, so it is a legitimate model of a mitochondrial translation disorder, although some caution is warranted as p32 is also found outside mitochondria.

Here the authors show that in the p32 KO, p62 and LC3B steady state levels increase in the heart and these proteins form inclusions (fig 1B, C). p62 accumulates around mitochondria, based on 2 foci (fig 1E), which is interpreted as arrested mitophagy. Several autophagy related genes are more highly expressed in the KO heart than control tissues. Taken together the conclusion that "autophagy is initiated but further steps is (sic) suppressed in the heart of p32cKO mice at nine months" is reasonably robust. Furthermore the authors present evidence of lysosomal differences/abnormalities compared to age-matched control tissue (accumulation of lipofuscin, and high levels of the lysosomal protein Lamp2 relative to VDAC (although the mitochondrial protein VDAC is a strange choice of reference, a better marker would be a cytoskeletal protein)). ULK is activated again consistent with an (attempted) upregulation of autophagy. Taken together these data add to earlier works that have linked mitochondrial and lysosomal dysfunction.

Next the authors carried out metabolite analysis and found NAD⁺ and NADP⁺ to be low in the p32 KO mouse heart. Gene expression analysis suggested this is due to decreased NAD synthesis. Based on a study in *Drosophila* the authors inferred the changes in gene expression stemmed from HIF1a induction. Others have demonstrated that HIF1a and OXPHOS are coupled, and here the authors show that HIF1a is upregulated in the p32 KO heart, and that this can be attributed to the mitochondrial translation/OXPHOS defect as chloramphenicol has the same effect on HIF1a, and on Nmam3 and Nmam2 expression (and that this effect is blocked by a HIF inhibitor). Therefore, this section is well reasoned and executed and provides a set of data that all support each other (except that fig 3F appears to be a single experiment and so needs to be shown to be reproducible).

The data showing that NAD rescues the lysosomal acidification problem (fig 4) rest on the assumption that lysotracker is an appropriate reporter. Later, the authors use LysoSensor DND-160 (fig 5) to show a nampt inhibitor decreases lysosomal acidification. LysoSensor DND-160 should be used to substantiate the claims made for fig 4. Nevertheless, the 'rescue' only with the nmnat3 variant lacking an MTS (fig 4) is a nice result.

Fig 5 seems to be drifting away from the main thrust of the manuscript, which is mitochondrial dysfunction and its effects on autophagy and NAD homeostasis. Hsu et al 2009 have previously shown that Nampt downregulation has similar effects to chloroquine, so it is expected that Namn KD would impact lysosomes/autophagy.

The hypothesis for GAPDH/PGK-dependent ATP production supporting lysosomal acidification is valid but the demonstration that some (loosely bound) GAPDH/PGK co-fractionates with lysosomes 6A-6C is insufficient to claim that: "These observations strongly suggest that ATP is produced by the lysosome bound GAPDH/PGK and more effectively supports vesicular proton uptake than free cytosolic ATP." That these enzymes are active is neither surprising nor informative (6E).

This reviewer recognizes the substantial amount of additional work involved in producing and analyzing the double conditional KO mouse (ATG5/p32). However, the results of the dcKO add little

of substance to the study.

In summary, the study adds to a growing body of work showing that mitochondrial dysfunction impacts other areas of cell biology and metabolism. A p32 KO with impaired mitochondrial translation/OXPHOS displays lysosome abnormalities stemming from disturbed NAD metabolism. Concordantly, lysosomal acidification is corrected by NMN supplementation in p32 KO MEFs. The study does seem to lose its way, towards the end and culminates in a major experiment that brought little reward. The obvious question of whether NMN or related compound is of benefit to the p32 KO heart is not addressed.

Other comments and questions:

As the authors note Yamada et al. found that p62 accumulated around mitochondria in Drp1KO hepatocytes and that it was co-localized with ubiquitin and LC3 (Yamada et al., 2018). Taken together with the current findings the implication is that various forms of mitochondrial dysfunction cause p62 accumulation and impede autophagy. This means the initial part of the study of the p32 KO did not yield novel findings.

The starting hypothesis is in one case weak. Mitochondria with "internal collapse" (i.e. - cristae loss) is a well understood feature of mitochondrial respiratory chain dysfunction. The mitochondria are larger because the lack of cristae junctions decreases the surface area of inner membrane that can 'fit into' a mitochondrion of a given volume). Hence, this is no reason to speculate that mitophagy is impaired or repressed.

TEM of Fig. 1A shows 1 abnormal mitochondrion in the KO. Many more sections need to be disclosed.

"The size of the p62 and LC3 ring staining pattern was similar to that of the giant mitochondria determined by electron microscopy analysis (Figure 1A), suggesting that LC3-positive autophagosomes sequester damaged mitochondria in p32cKO heart." More examples of each juxtaposed are needed to justify this conclusion.

"fractionation of mitochondria and lysosomes from the membrane fraction demonstrated the Nmnat3 expression in both lysosomal and mitochondrial fractions (Figure 3D)." Expression is not the correct word here, re-write.

The nmnat2 data in fig 4B,C should move to supplemental data, as it is better to lead with nmnat3 (as it is more relevant to the heart and the data are better).

Introduction:

The introduction is fragmented, the subject switches from p32 to NAD to lysosomes without any links.

The end of the Introduction needs re-writing the final paragraph describes ATG5 deficient mice that are a minor part of this study and ends with a sweeping (ungrammatical) statement that lacks context and alludes to but does not describe the important findings. "In the present study, we explain why mitochondrial dysfunction induced loss of lysosomal function, leading to the

autolysosome defect, and provide new insight into the essential nature of NAD⁺ in normal lysosomal function."

"It is unclear why cardiomyocyte-specific p32-deficient mice survived for a long time". - this statement is unnecessary and uninformative, better to report the facts and the clear conclusions. The results indicate that the mice can survive with cardiomyopathy for 10 months.

Fig S2 translation (typo)

Referee #1:

Reviewers' Comments to Author
EMBOJ-2020-105268

Previous Manuscript #:

We sincerely appreciate that you provided several critical comments. We are sure that your suggestions were very important for improvement of our manuscript. We added several experiments and extensively improve this new re-submitted manuscript according to your comments.

1. The concept is very interesting, and centered on the nascent field of mitochondria-lysosome crosstalk. However, I have several concerns related to novelty and methodology.

According to reviewer's comment, we added the sentence in Introduction in this resubmitted paper.

“Recently, mitochondrial dysfunction has been shown to have pleiotropic effects in multicellular organisms, affecting other organelles such as lysosomes. Mitochondrial respiration deficiency impairs lysosomal function, promotes p62 and sphingomyelin accumulation and disrupts endolysosomal trafficking pathways and autophagy, thereby linking primary mitochondrial dysfunction to a lysosomal storage disorder (Baixauli et al, 2015). It has also been reported that mitochondrial respiratory chain deficiency inhibits lysosomal hydrolysis and lysosomal function, and the relationship between mitochondria and lysosomal function was described via various pathways (Demers-Lamarche et al, 2016; Deus et al, 2020; Fernandez-Mosquera et al, 2017; Fernandez-Mosquera et al, 2019). Thus, the crosstalk between mitochondria and lysosomal function has been revealed.”

Novelty

1. HIF1 α , which is stabilized by mitochondrial translation dysfunction, suppressed Nmnat3 gene expression, suggesting that the HIF1 α -Nmnat3-mediated NAD⁺ production
2. The glycolytic enzymes, GAPDH/PGK, were associated with lysosomal vesicles and NAD⁺ was required for ATP production around lysosomal vesicles.

2. Methodological detail

My methodological concerns are detailed below, but they can be summarized in few words: the method that the authors used to obtain lysosomal fractions is not

adequate for this particular study because it is not possible to exclude the possibility that there are cytoplasmic contaminants. This is a fundamental flaw, given that the model proposed here posits that it's two cytoplasmic proteins that attach to the lysosome that are involved in the synthesis of the ATP that is needed for vATPase function. The last experiment is made with magnetic-purified lysosomes - the others need to be as well.

According to reviewer's comment, we performed three different methods by purification of lysosome in this resubmitted paper.

Please see the result section "Lysosomal fractions contain GAPDH, PGK and NAD⁺, which are linked to ATP production around lysosomes."

We also added another IP method by using the TMEM192-HA, then we observed that few cytosol contamination.

3.. Finally, two experiments are, in my view, missing to prove the model that the authors propose. If the problem of the vATPase is that there is not enough ATP, why can't it take ATP from the cytoplasm? Does this also happen in high-glucose medium, i.e., is it that there is not enough ATP production in this low glucose-medium? This is a fairly simple experiment and would be informative.

According to the comments, we performed the experiment and described in Result section. "To investigate the importance of local ATP production surrounding lysosomes, we investigated the acidification and function of lysosomes in high-glucose medium. High glucose provides the cytoplasm with a sufficient amount of ATP. High-glucose medium did not affect lysosomal function, whereas FK866-dependent NAD depletion reduced lysosomal function (Fig EV5G), suggesting that ATP was not supplied by high glucose-dependent glycolytic ATP production."

In Discussion section, we described that

"We showed that WT MEFs were highly dependent on mitochondrial ATP production, and p32KO MEFs were dependent on glycolytic ATP production (Yagi et al., 2012). p32KO MEFs showed less lysosomal activity and the total ATP levels were significantly higher in p32KO MEFs than in WT MEFs, suggesting that the difference in ATP production by mitochondria and glycolysis was not involved in lysosomal function. Therefore, local ATP production may be important for lysosomal function."

4.. The second has to do with the Nmnat3 that the authors propose locates to the

lysosome (my detailed discussion about the problem with that experiment is below): what makes it go to the lysosome? A study has shown that lysosomal malfunction caused by mitochondria is due to lower PI(3,5)P2 in the lysosome. Could this affect the recruitment of Nmat3, or of the glycolytic proteins, to the lysosome? The authors could easily test if inhibition of PIKfyve, the only enzyme that makes PI(3,5)P2 in the lysosomal membrane, would affect the localization of the glycolytic enzymes or of Nmat3 (or the expression of Nmat3).

According to the review's comment, we performed the experiment and describe in Result section. In this resubmitted paper, we did not describe whether Nmnat3 is localized in lysosomal fraction because of unstable expression of Nmnat3 in MEF cells. We focus on localization of glycolytic enzyme such as GAPDH/PGK to lysosome.

"Next, we investigated whether inhibition of PIKfyve (YM201636), the only enzyme that catalyzes PI(3,5)P2 production in the lysosomal membrane (Bissig et al, 2017), affects the localization of the glycolytic enzymes, GAPDH/PGK. The addition of inhibitors did not significantly change the amount of glycolytic enzymes localized to lysosomes (Fig 6G), suggesting that PI(3,5)P2 does not affect the transport of glycolytic enzymes to lysosomes."

5.. "When cells are deprived of nutrients, to maintain metabolic homeostasis and cell viability, mitochondria can also be degraded non-selectively together with other cytosolic contents (Youle & Narendra, 2011)." - In fact, mitochondria are known to be protected from autophagy during starvation, via hyperfusion (PMID 21478857, PMID 21646527).

We agree the review's comment, we deleted the above sentence in Introduction section.

"When autophagy is triggered, mitochondria elongate in vitro and in vivo. During starvation, cellular cyclic AMP levels increase and protein kinase A (PKA) is activated. PKA in turn phosphorylates the pro-fission dynamin-related protein 1 (DRP1), which is therefore retained in the cytoplasm, leading to unopposed mitochondrial fusion"

In our mouse models, mitochondria translation deficiency heart showed the elongated mitochondria due to above hypothesis. Then, we added the sentence in Result section.

"The size of the p62 and LC3 ring staining pattern was similar to that of the giant mitochondria determined by electron microscopy analysis (Fig 1A), suggesting that LC3-positive ring sequester damaged mitochondria in the p32cKO heart. When

autophagy is triggered, mitochondria elongate in vitro and in vivo (Rambold et al, 2011), suggesting that mitochondrial translation deficiency heart show the elongated mitochondria.”

6.. "no clear relationship between mitochondrial dysfunction and lysosomal function has been revealed." - this is simply not true. The interdependence between mitochondria and lysosomes, and particularly how dysfunctional mitochondria perturb lysosomal function has been studied by several labs (PMID 26299452, 26987902, 30917721, 28345620, 31791731), and I don't see how the authors chose not to incorporate this literature into the manuscript.

According to review's comments, we delete the above sentence, then we changed the new sentence in Introduction in this resubmitted version.

“Recently, mitochondrial dysfunction has been shown to have pleiotropic effects in multicellular organisms, affecting other organelles such as lysosomes. Mitochondrial respiration deficiency impairs lysosomal function, promotes p62 and sphingomyelin accumulation and disrupts endolysosomal trafficking pathways and autophagy, thereby linking primary mitochondrial dysfunction to a lysosomal storage disorder (Baixauli et al, 2015). It has also been reported that mitochondrial respiratory chain deficiency inhibits lysosomal hydrolysis and lysosomal function, and the relationship between mitochondria and lysosomal function was described via various pathways (Demers-Lamarche et al, 2016; Deus et al, 2020; Fernandez-Mosquera et al, 2017; Fernandez-Mosquera et al, 2019). Thus, the crosstalk between mitochondria and lysosomal function has been revealed.”

7.. "The expressions of autophagy-related *Gabarap11*, *Atrogin1*, *Lamp2*, and *Atg4b* genes were increased in the heart of p32cKO mice compared with that in WT mice (Figure 1G)." - the figure shows that *Atrogin1* is down, the others are up. The text should be corrected accordingly.

According to review's comments, we changed the sentence in this resubmitted paper.

“The expression of the autophagy-related genes, *Gabarap11*, *Lamp2* and *Atg4b*, was increased, whereas that of the muscle-specific ubiquitin ligase, *Atrogin1*, was decreased in the heart of p32cKO mice compared with that in the heart of WT mice (Fig EV1E). It has been reported that *Atrogin1* deficiency promotes cardiomyopathy and premature death via impaired autophagy (Zaglia et al, 2014). Consequently, the autophagy-related

molecules are gathered around the damaged mitochondria as a result of autophagy suppression.”

8.. Figure 2B - autofluorescence appearance? Number of autofluorescent particles or...?

We change the sentence in this resubmitted paper.

“Broad-spectrum autofluorescence is a characteristic property of lipofuscin. In the p32cKO heart, the number of autofluorescing particles increased around the nuclei over time, whereas in the WT heart it did not (Fig 2B and Fig EV1F). Collectively lysosomes in p32cKO hearts seemed morphologically impaired.”

9.. "Collectively lysosomes and autolysosome in p32cKO hearts seem functionally impaired." - from figures 2A-2C, we can conclude that the lysosomes show morphological signs of dysfunction. To conclude that they are functionally impaired more experiments are required, i.e., which aspect of lysosomal function is actually impaired.

According to review’s comments, we change the sentence in resubmitted paper.

“Collectively lysosomes and autolysosome in p32cKO hearts seem morphologically impaired. “

10.. Figure 2E - why VDAC (mitochondrial outer membrane) as a loading control for a whole cell extract? Especially given that the p32-KO cells have a mitochondrial defect...

According to review’s comments, we changed the sentence in this resubmitted paper

We changed the loading control to GAPDH instead of VDAC in this Figure 2D in resubmitted paper.

11.. Figure 2G - the representative western for ULK1-555P does not match the quantification

"The phosphorylation of ULK1 Ser757 (moles per mole) from p32cKO heart was increased 4-fold compared with WT samples, in agreement with the overall increase in mTOR signaling (Figure 2G)." - there is no assessment of mTORC1 activity (the mTOR complex that regulates ULK1 is the mTORC1, so that rather than mTOR should be mentioned. Furthermore, in the study that shows the effect of p32 on mitochondria (Saito et al, Cardiovascular Research,2017, 113: 1173-1185), the authors conclude that mTORC1 is inhibited. Therefore, it is important that the authors clarify what happens to

mTORC1, and if mTORC1 is down, why would ULK1 be more phosphorylated in the site targeted by mTORC1 (Ser757).

According to review's comments, we change the sentence in this resubmitted paper.

“Next, we examined the phosphorylation of the autophagy-related proteins, p62 and ULK1, and the progress of autophagy. Ser351 phosphorylation of p62 was not altered in p32cKO compared with WT samples, but Ser403 phosphorylation, which is induced by ULK1, was clearly increased (Fig 1G). Furthermore, the phosphorylation of ULK1 Ser757 in the p32cKO heart was increased 4-fold compared with WT samples (Fig EV1D), which is consistent with the autophagy initiation process by LC3-II ring formation. It has been suggested that AMPK activated autophagy via two independent mechanisms: suppression of mammalian target of rapamycin complex 1 (mTORC1) activity and direct control of ULK1 phosphorylation (Zhao & Klionsky, 2011). Previously, we have observed AMPK α phosphorylation in p32cKO hearts (Saito et al., 2017), which seems to initiate autophagy, but the subsequent steps are suppressed in the heart of p32cKO mice.”

We also added the sentence in Discussion

“In our mouse model, we have previously observed increased AMPK α T172 and 4E-BP1 phosphorylation and decreased S6K phosphorylation, suggesting that the mTOR pathway is inhibited in p32-KO heart (Yagi et al., 2012). These differences are due to differential mitochondrial dysfunction between the respiratory chain and mitochondrial translation. We also observed that mitochondrial translational inhibition induced HIF1 α stability and several genes involved in glycolysis or NAD⁺ synthesis genes and, finally, decreased lysosomal acidification (Fig 3E–I). These results suggest that mitochondrial-lysosomal interactions use several signaling pathways that are mechanistically distinct.”

12.. "Taken together, we consider that the degradation mechanism by autophagy was rather more strongly initiated but stopped midway" - this conclusion is at odds with the data. The ULK1 phosphorylation at Ser757 is inhibitory of the formation of autophagosomes at a very early stage. So it cannot be concluded that autophagy was strongly initiated.. The "stopped midway" likely refers to a stalling of the autophagic flux, which is likely caused by the lysosomal impairment. This should be rewritten to fit what the data shows.

According to review's comments, we change the sentence in this resubmitted paper.

“Taken together, we speculate that the degradation mechanism by autophagy was stopped prematurely and impaired lysosomes were involved in autophagic dysfunction.”

13.. "To know which Nmnat isozyme is responsible, we examined the expression and localization of Nmnat2 and Nmnat3 in cardiac tissue by western blotting." - the authors should clarify why Nmnat1 was not examined.

According to review's comments, we change the sentence in this resubmitted paper.

“To elucidate which Nmnat isozyme is responsible for this process, we examined the expression and localization of cytoplasmic Nmnat2 and Nmnat3 in cardiac tissue, because Nmnat1 is mainly localized in the nucleus (Fig EV2A).”

14.. Figure 3D - the method used is rather unclean. There are much cleaner and accurate methods established that allow the dissection of lysosomal and mitochondrial proteins fairly easily. Given that this is an important point that the rest of the study builds on, I urge the authors to prepare clean mitochondrial and lysosomal preps (for example, via immunoisolation, see PMID), and show that Nmnat3 is present in the organelle preps. As such, the method does not allow for an unequivocal interpretation. A similar concern regarding Figure S1B, with the added question that if the authors want to build on cytoplasmic localization of Nmnat3, then the western that they show also shows leakage of VDAC, which is an outer mitochondrial membrane protein. This suggests that there was contamination of the cytoplasmic prep with mitochondria, and begs for the levels of some soluble mitochondrial protein to determine the level of leakage. (also, important to distinguish cytoplasmic soluble versus lysosomal)

According to review's comments, we change the sentence in this resubmitted paper. We described that localization of nmnat3 is cytoplasm (Cytosol and mitochondria) in mouse heart.

“We isolated the membrane and cytosol fraction from WT heart (Fig EV2B). Consistent with a previous report that Nmnat2 is mainly expressed in the brain (Ali et al, 2013), no protein expression was detected in the heart tissue (Fig EV2C). Nmnat3 was detected in the p32cKO heart (Fig 3C) and was thought to localize in mitochondria. However, recently its localization in the cytosol has been reported (Yamamoto et al, 2016). In fact, abundant Nmnat3 was also found in the cytosol of the heart (Fig EV2C). These results suggest that Nmnat3 is expressed in the cytosol and the membrane fraction of the heart

and contributes to NAD⁺ synthesis in the cytosol.”

15.. Figure 3F - there are no error bars or statistical treatment, not possible to make a conclusion on the result

According to review’s comments, we perform the triplicated experience and change the Figure 3G and 3I in this resubmitted paper.

16.. "To examine lysosomal acidification, we used LysoTracker, a qualitative acidic pH indicator dye that is enriched and strongly fluoresces in compartments with a pH below 5 (Lee, Yu et al., 2010). LysoTracker staining was weak in p32-knockout MEFs compared with WT cells, suggesting that lysosomal acidification was reduced in p32-knockout MEFs (Figure 4A)." - as the authors correctly point out, LysoTracker is only a qualitative indicator. The biggest issue here is that there is no control for the amount of lysotracker that is loaded to the lysosome. So the signal observed may be a reflection of higher pH or of lower uptake by the lysosome, or both. A more quantitative method should be employed, such as the green/red dextran approach (two molecules of the same size, to control for delivery, and two different fluorophores, one pH-sensitive and the other -insensitive; the ratio reflects solely the luminal pH). Furthermore, the authors fixed the cells after lysotracker treatment, which is not indicated as it affects the results. Finally, I cannot find in the methods or figure legend how long was NMN added before imaging - it should be added (and how much, and what was the vehicle). This is also a concern for the other panels of Figure 4 that rely on lysotracker, as well as Fig S3.

According to review’s comments, we used the recommended probe in this resubmitted paper. (Fig 4A , Fig 5A and Fig EV3A)

“ To examine lysosomal acidification, we used two fluorescently tagged probes: the pH-sensitive Oregon green 488–dextran and the pH-insensitive tetramethyl rhodamine-dextran. Oregon green 488 has a pKa of 4.7, which is suitable for measuring the acidic pH of the lysosomal lumen.” We also change the legend in Fig EV3A.”

17.. Figure 4B - the overexpression in the WT cells should also be shown, inclusive on the pH experiments, so that it can be assessed if the rescue of the KO brings it to WT levels or somewhere in between - and overexpression of NMNAT affects the WT.

We observed that “Addition of NMN to p32KO MEFs restored the lysosomal

acidification (Fig 4A). Furthermore, we observed less acidification upon CAP treatment, suggesting that mitochondrial translation deficiency affects lysosomal acidification (Fig 4A). After addition of NMN to WT MEF, there are no less acidification, then, we did not perform whether overexpression of *nmnat3* affect the acidification of WT MEF cell in Fig 3B-3D.

18.. Figure 5 - the images have too much staining background to allow for proper colocalization quantification. It is important that the authors show the individual channels as well, so that one can assess the LAMP2 staining (very clear on the control, but impossible to assess on the others). Once the individual channels are presented, it will be possible to have a better overview of this result.

According to review's comments, we change the individual channels of Figure 5E, 5F in this resubmitted paper.

19.. "After bafilomycin A treatment, an increase in DALGreen staining as well as fusion of DAPRed and DALGreen staining was observed, suggesting a reduction in autophagic degradation within autolysosomes (Figure 5D)." - this is a puzzling observation. The treatment with bafilomycin is expected to neutralize the lysosomal pH, so why would DALGreen, which fluoresces more in acidic pH, be enhanced? As mentioned before, the duration, concentration and vehicles of the treatments have to be indicated. It is hard to assess this experiment from a steady-state point-of-view. If the effect of the FK866 is fast (and given that NAD⁺ is synthesized and used recurrently, it is expectable that the effect is fast), it would be much more informative to show this experiment using live imaging, and present the videos or a time-course. (I read now figure S5, in which the timeframe for FK866 treatment is indicated as 24-48h. This is quite a long treatment. How long after the FK866 is added to the cells do the concentrations of NAD⁺ start decreasing?)

Thank you for comment. In previous paper, we discussed that the number of autophagosome and autolysosomal staining were increased after nutrient starvation, then, we described that "After bafilomycin A treatment, an increase in DALGreen staining as well as fusion of DAPRed and DALGreen staining was observed, suggesting a reduction in autophagic degradation within autolysosomes (Figure 5D)". in previously paper. It means that increase is the number of DALGreen staining, not intensity of DALGreen.

In this resubmitted paper, we changed the sentence in this resubmitted paper.

“After bafilomycin A treatment, an increased number of fusion of DAPRed and DALGreen staining was observed, suggesting a reduction in autophagic degradation within autolysosomes (Fig 5F). We also found increased number of fusion of DAPRed and DALGreen staining after FK866 treatment, suggesting that reduced NAD^+ levels inhibited autophagic degradation because of lysosomal dysfunction followed by accumulation of both autophagosomes and autolysosomes (Fig 5F).”

20.. "These results indicate again that inhibition of NAD^+ synthesis impairs lysosomal function and resultantly suppresses degradation by autolysosome" - to make statements on lysosomal degradation, it is necessary to assess it in more detail, by monitoring for example cathepsin activities with Magic Red.

According to the review's comments, we performed the lysosomal proteolytic activity by using polymeric bovine serum albumin labeled with a green BODIPY dye (DQ Green BSA green; the assay is based on the fact that the high amount of fluorophore in polymeric DQ Green BSA has a quenching effect).

In this resubmitted paper, we added the new result.

“Compared with the control, FK866-treated WT MEFs had a lower rate of DQ Green BSA hydrolysis, suggesting a decrease in the activity of lysosomal proteases, which indicates that lysosomal function requires NAD^+ synthesis (Fig 5C). CAP-treated MEFs and p32KO MEFs also showed reduced lysosomal degradation activity in a time-dependent manner, suggesting that mitochondrial translation deficiency affected the lysosomal degradation activity (Fig 5D and Fig EV4B).”

21.. Figure 6 - cleaner method to isolate lysosomes needed, as mentioned above. "p62 was also found in the lysosomal fraction, suggesting that fractions 2 and 3 represent the lysosome and autolysosome fractions" - or other structures, there is no way to be sure that this is only lysosomes. That is why it is pivotal to make a pure lysosomal prep.

According to review's comments, we deleted the p62 in this resubmitted paper.

22.. "The analysis of these subcellular fractions indicated that GAPDH and PGK are in the cytosol and can bind to various types of membranes such as lysosomal membranes." - or that there are cytoplasmic contaminants in all fractions.

According to the review's comments, we change the sentence in Results section.

"We first isolated the lysosomal fraction, and then examined whether GAPDH and PGK are localized in this fraction. To purify lysosomal fractions, the membrane fraction was washed twice to remove the free cytosolic proteins, and then adjusted with a discontinuous OptiPrepTM density gradient (Fig 6A). After ultracentrifugation, five individual fractions were collected from the top of the gradient, and then analyzed for the presence of characteristic organelle marker proteins (Fig 6B).

Lamp2 and V-ATPase were enriched in fraction 2, suggesting lysosomal fraction. Voltage-dependent anion channel (VDAC) and the glycolytic enzyme, Hexokinase 2 (HK2), were not found in the lysosomal fraction but were detected in the mitochondrial fraction (fractions 4, 5). After two washes and centrifugations, the membrane fraction was concentrated, and very little protein was detected in fraction 2, suggesting that this method results in little cytosolic protein contamination (Fig EV5A and EV5B). Furthermore, mTOR protein and the lysosomal V-ATPase were enriched in fraction 2, whereas no Hsp70 protein was observed, suggesting that there was little contamination of free cytosolic proteins in fraction 2. We found that PGK was only present in the lysosomal fractions (fractions 2, 3) and GAPDH was present in all fraction because GAPDH is localized in specific subcellular compartments including fusion vesicles by palmitoylation."

23.. "We next evaluated whether GAPDH and PGK are in a complex in the cell by co-immunoprecipitation experiments. HeLa cells transfected with a GAPDH-HA plasmid were lysed" - why the change from MEFs to HeLa cells for this particular experiment? MEFs were transfected before in the paper, so transfection shouldn't be an issue. Furthermore, the fact that the IP is performed using overexpression of one of the partners, begs for the reverse IP. Both should be done in MEFs or cardiomyocytes, to ensure that the system is operative in the same cells where the phenotype was described.

According to review's comments, we performed IP with MEF cells and change the Fig 6C in this resubmitted paper.

24.. "We incubated the lysosomal vesicles fraction with GAP, ADP, Pi, and NAD to measure ATP production. We observed significant GAP-dependent ATP production under these conditions (Figure 6E), showing that both GAPDH and the downstream PGK are enzymatically active in the lysosomal fraction" - this goes back to the previous point in which it was not possible to assess if the presence of GAPDH and PGK was

due to cytoplasmic contamination of fractions containing lysosomal components. The preparation of pure lysosomes is fundamental for the conclusions. For example, using the magnetic bead strategy employed in 6F.

According to review's comments, we tried the 2 different method in this system. "This separation study strongly suggests that PGK and GAPDH are physically associated with lysosome."

"GAPDH and PGK may exist in a complex (Fig 6C). To determine the nature of the interaction of GAPDH and PGK with lysosomal vesicles, lysosomal vesicles were treated with various concentrations of NaCl. PGK and GAPDH dissociated from lysosomal membranes with increasing salt concentrations (Fig 6D), suggesting that GAPDH and PGK are associated with lysosomal vesicle membranes peripherally by ionic bonds. At 0.8 M NaCl, GAPDH and PGK were found to remain associated with lysosomal vesicle membranes, suggesting that GAPDH and PGK are strongly associated with lysosomal vesicle membranes."

25.. "These results suggest that mitochondrial dysfunction due to mitochondrial translation deficiency is not directly involved in autophagy and severe heart failure. Both of the ATG5cKO mice and ATG5/p32 dcKO mice died at seven months, suggesting that autophagy failure was a major cause of p32cKO mice death (Figure 7H)." - It has been shown that lysosomal malfunction triggered by defects in mitochondria is not due to autophagosome delivery to the lysosomes (PMID). This experiment should therefore be placed in the context of published literature.

According to review's comments, we deleted the analysis of p32/ATG5 double knockout mouse in this resubmitted paper.

26.. "Finally, these mice suggest that autophagy dysfunction due to mitochondrial dysfunction in hearts leads to severe heart failure and death." - this sentence is at odds with the previous sentence, and should be corrected.

According to review's comments, we delete the p32/ATG5 double knockout mice, then, we deleted the sentence in this resubmitted paper.

27.. The DRP1 chapter on the discussion is out of place and doesn't really make sense with the rest of the manuscript.

Once more, these findings should be placed in the context of the published literature.

According to review's comments, we deleted the Drp1 section in Discussion in this resubmitted paper.

Referee #2:

Previous Manuscript #: EMBOJ-2020-105268

In the manuscript "Mitochondria translation deficiency impairs NAD⁺ mediated lysosomal acidification and autophagy" the author describes the physiological role of p32 in cardiomyocytes. The manuscript is composed of several parts. At the beginning the authors described the role of p32 in heart. The absence of p32 determines a phenotype with the presence of abnormal mitochondria and lysosome. These two features could be explained with a problem in the autophagy flux. A second part is dedicated to the role of NAD⁺ in p32 deficient cells and in particular the authors focus on Nmnat3 gene expression and positive function of the cytosolic form of this gene. They also investigate the beneficial effect of FK866 on lysosomes and autophagy. A third part is more on the molecular mechanisms and in particular regarding the ATP production on lysosomal surface. The last part looks back to the autophagy impairment. The manuscript has interesting points; therefore, the huge amount of information sometime is more confusing than clarifying. It is difficult to follow what the authors really want to say and what is the final message. Many scientific questions have been opened but only few of them have been really addressed in details. There are still points that need to be properly addressed in terms of experimental data and discussion. Moreover, the impairment of autophagy, that leads to mitochondria accumulation, is a well-known concept. The fact that p32 knockout negatively affects autophagy flux and the mice hearts present abnormal mitochondria is quite awaited.

We sincerely appreciate that you provided several critical comments. We are sure that your suggestions were very important for improvement of our manuscript. We added several experiments and extensively improve this new re-submitted manuscript according to your comments.

According to review's comments, we delete the analysis of double knockout mouse in this manuscript and we focus on the novelty

1. HIF1 α , which is stabilized by mitochondrial translation dysfunction, suppressed Nmnat3 gene expression, suggesting that the HIF1 α -Nmnat3-mediated NAD⁺ production
2. The glycolytic enzymes, GAPDH/PGK, were associated with lysosomal vesicles and NAD⁺ was required for ATP production around lysosomal vesicles.

Major points:

- 1.. Figure 1: the fact that mitochondria are surrounded by Ub, p62 and LC3 is not sufficient to state that they are inside autophagosomes. Electron microscopy imagines in

the heart tissue or cytofluorimetry assay in cardiomyocytes using mKeima and/or RFP-GFP probes linked to mitochondria proteins would be more informative.

According to review's comments, we change the sentence in Result section, "indicating that mitochondria are surrounded by autophagy related protein. The size of the p62 and LC3 ring staining pattern was similar to that of the giant mitochondria determined by electron microscopy analysis (Fig 1A), suggesting that LC3-positive ring sequester damaged mitochondria in the p32cKO heart."

2.. Page 8 paragraph "Lysosomal morphology is impaired in the heart of p32-knockout mice". This title is a bit misleading. Morphology could be altered or abnormal, function can be impaired. Both, lysosomal morphology and function, seem to be affected in p32-knockout, however, more experimental evidences are needed.

According to review's comments, we change the sentence in this paragraph.

We found that lysosomal morphology is altered in p32cKO heart, then "we speculate that the degradation mechanism by autophagy was stopped prematurely and impaired lysosomal morphology might be involved in autophagic dysfunction."

In vitro by using the DQ Green BSA, we demonstrated that "CAP-treated MEFs and p32KO MEFs also showed reduced lysosomal degradation activity in a time-dependent manner, suggesting that mitochondrial translation deficiency affected the lysosomal degradation activity (Fig 5D and Fig EV4B)."

"Compared with the control, FK866-treated WT MEFs had a lower rate of DQ Green BSA hydrolysis, suggesting a decrease in the activity of lysosomal proteases, which indicates that lysosomal function requires NAD⁺ synthesis (Fig 5C). "

These result suggested that mitochondrial translation deficiency affect lysosomal function.

3.. Figure 2: abnormal lysosomes are clear. What should be described more into details is the presence of mitochondria inside autolysosomes. Fig 2C is not really informative. Moreover, to claim that there is an autophagy flux problem, which likely is present, author should provide additional autophagy flux assays in order to exactly point where the problem is.

According to review's comments, we deleted suggesting points which autophagic flux are stopped. In this manuscript, we focus on lysosomal dysfunction in p32 deficiency

heart.

We also deleted the Fig 2C (the presence of mitochondrial inside autophagosome.)

4.. The NAD⁺ regulation and its role in lysosomal acidification is interesting. Data presented indicate that p32 has an effect on Nmnat3 and on lysosomes too. This part is interesting therefore few more experiments oriented to better elucidate the signaling pathway p32-HIF1alpha-Nmnat3.

According to review's comments, we performed another experience and change the sentence and added the Fig 3F-3I in this manuscript.

“We found that HIF1 α was significantly upregulated in the p32cKO heart compared with the WT heart (Fig 3E). Because p32 is involved in mitochondrial translation, we used antibiotics such as chloramphenicol (CAP) to inhibit mitochondrial translation. CAP induced HIF1 α expression in WT mouse embryonic fibroblasts (MEFs) in a time-dependent manner (Fig 3F). Moreover, CAP treatment reduced Nmnat3 expression (Fig 3G). We also observed that CoCl₂ treatment, which stably induces HIF1 α expression, suppressed Nmnat3 gene expression (Fig 3H-I). A chromatin immunoprecipitation (ChIP) database analysis (ChIP-Atlas: <http://chip-atlas.org/>) showed that HIF1 α can associate with promoter regions of the Nampt, Nmnat1–3 and Sirt3 genes in several cell lines (Fig EV2D). In contrast, a HIF1 α inhibitor suppressed the CAP inhibitory effect on Nmnat3 expression (Fig 3G), suggesting that mitochondrial translation inhibition induced HIF1 α expression, leading to suppression of Nmnat3 expression. “

5.. The use of FK866 is of potential interest as a modulator of lysosomal function and autophagy flux. Therefore, more specific and detailed experiments are needed to support that lysosomal activity and autophagy flux have been restored. The reported data are not sufficient to make solid conclusions.

According to review's comments, we performed the functional assay by using the green BODIPY dye (DQ Green BSA)

In this manuscript, we added the sentence in Result and Fig 5C and 5D.

“To assess if the increased pH affects lysosomal proteolytic activity, we incubated the FK866-treated MEFs with polymeric bovine serum albumin labeled with a green BODIPY dye (DQ Green BSA; the assay is based on the fact that the high amount of fluorophore in polymeric DQ Green BSA has a quenching effect). DQ Green BSA

enters the cell via endocytosis and accumulates in the lysosomes (Marwaha & Sharma, 2017). Because the lysosomal proteases hydrolyze DQ Green BSA into small peptides, the fluorescence is no longer quenched, fluorescent monomers accumulate and an increase in fluorescence is observed. Moreover, the rate of this fluorescence increase is proportional to the activity of the lysosomal proteases.

6.. The paragraph regarding the molecular mechanisms linking GAPDH, PGK, NAD⁺ and ATP production at the lysosome surface is interesting and actually point to the role of the lysosome as signaling hub. This part deserves a more critical analysis in terms of experimental data. Conclusions are interesting but not fully supported by data.

According to review's comments, we performed 3 different method to purify the lysosome fraction and show GAPDH/PGK were tightly associated with lysosome and the NAD⁺ is essential for ATP production near the lysosome in Fig 6.

7.. Last paragraph pages 20-21 can be fused together with the first one. In both cases the authors are describing impaired autophagy phenotype.

According to review's comments, we change the sentence in this manuscript.

8.. Discussion is too long and lengthy. It is difficult to follow what the authors want to point out. I would suggest to significantly short the text and clearly focus on the main message of the manuscript.

According to review's comments, we change the discussion in this resubmitted paper.

9.. The manuscript has potential but it is difficult to catch the message that the authors want to deliver. In order to be considered for publication the manuscript should be reorganized to be simpler to read and more focused to the final message. Several considerations regarding the role of autophagy in heart cells could be shorted and data moved to supplementary. Autophagy in heart has been investigated in the past. Mitochondria translation deficiency, that is the pointed in the title, should have a more relevant position in the manuscript and more experimental data in its support.

The crosstalk between mitochondria and lysosome is also interesting and it could be highlighted and studied deeper.

According to review's comments,

“We focus the point concerning the mitochondrial -lysosomal interaction as NAD

supplementation around the lysosome in this resubmitted paper “

We deleted the analysis of double knockout mouse in this manuscript.

Referee #3:

Previous Manuscript #: EMBOJ-2020-105268

We sincerely appreciate that you provided several critical comments. We are sure that your suggestions were very important for improvement of our manuscript. We added several experiments and extensively improve this new re-submitted manuscript according to your comments.

1.. This study sets out to investigate how autophagy is impacted by defective mitochondrial translation using a p32 KO mouse model. The mouse has previously been used to show that loss of p32 causes a severe mitochondrial translation defect and respiratory chain deficiency, so it is a legitimate model of a mitochondrial translation disorder, although some caution is warranted as p32 is also found outside mitochondria.

Thank you for the comment, we agree the p32 localization outside mitochondria and outside plasma in some paper. Especially p32 is localized in cell surface in cancer cell line, B lymphocytes or some inflammatory cells and also nuclear localization of tumor tissue at progression stage. However, in heart, p32 is mainly localized in mitochondria by immunohistochemistry and western blotting from fractionation of heart tissue (lane 5). In this paper, we assume that p32 localized to mouse heart mitochondria and p32 defect mainly induced mitochondrial translation deficiency.

2.. (although the mitochondrial protein VDAC is a strange choice of reference, a better marker would be a cytoskeletal protein)).

(except that fig 3F appears to be a single experiment and so needs to be shown to be reproducible).

According to review's comments, we changed the another marker protein GAPDH or β actin as control in Fig 3 and we performed the triplicated experience and show the SD bar (Fig 3G and 3I) in this resubmitted paper.

3.. The data showing that NAD rescues the lysosomal acidification problem (fig 4) rest on the assumption that lysotracker is an appropriate reporter. Later, the authors use LysoSensor DND-160 (fig 5) to show a nampt inhibitor decreases lysosomal acidification. LysoSensor DND-160 should be used to substantiate the claims made for

fig 4. Nevertheless, the 'rescue' only with the nmnat3 variant lacking an MTS (fig 4) is a nice result.

According to review's comments, we used two fluorescently tagged probes: the pH-sensitive Oregon green 488-dextran and the pH-insensitive tetramethyl rhodamine-dextran in Fig 4A and Fig 5A.

“To examine lysosomal acidification, we used two fluorescently tagged probes: the pH-sensitive Oregon green 488-dextran and the pH-insensitive tetramethyl rhodamine-dextran. Oregon green 488 has a pKa of 4.7, which is well suitable for measuring the acidic pH of the lysosome lumen. The emission was separately determined in individual lysosomes, and the fluorescence ratio was measured, allowing changes in lysosome pH to be monitored (more acidic show lower green/red ratio, on the other hand, less acidic show higher green/red) (Johnson, Ostrowski et al., 2016). “ in Result section.

4.. Fig 5 seems to be drifting away from the main thrust of the manuscript, which is mitochondrial dysfunction and its effects on autophagy and NAD homeostasis. Hsu et al 2009 have previously shown that Nampt downregulation has similar effects to chloroquine, so it is expected that Namn KD would impact lysosomes/autophagy.

According to review's comments, we observed the

“Treatment of WT MEFs with FK866, a Nampt inhibitor, depleted intracellular NAD⁺ and NADH (Fig EV4A) and reduced lysosomal acidification (Fig 5A). Because NMN is a product of the enzymatic reaction of Nampt, NMN was anticipated to restore NAD⁺ levels and rescue lysosomal function in the presence of FK866. Addition of NMN to FK866-treated cells restored the lysosomal acidification (Fig 5A) as assessed by dextran-488 and dextran-TMR, indicating that lysosomal function requires NAD⁺ synthesis. We also estimated lysosomal acidification using another probe, LysoSensor DND-160. The intralysosomal pH measured by this probe increased upon FK866 addition and this lysosomal pH increase was rescued by NMN (Fig 5B). “

Previously Hsu et al observed that Nampt downregulation inhibits autophagic flux and showed that Chloroquine treatment alone increased LC3-II and Ad-shRNA-Nampt failed to further increase LC3-II, suggesting that downregulation of Nampt does not increase autophagosome formation. These data suggested that they did not demonstrated the mechanism of lysosomal dysfunction after NAD depletion. In this paper, we first

showed that NAD is essential for lysosomal function through GAPDH/PGK near lysosome to maintain the lysosomal function.

In culture cell line, Nmnat3 is not expressed stably, then, we performed overexpression of Nmnat3, but fail to Nmnat3 knockdown experience in our system.

5.. The hypothesis for GAPDH/PGK-dependent ATP production supporting lysosomal acidification is valid but the demonstration that some (loosely bound) GAPDH/PGK co-fractionates with lysosomes 6A-6C is insufficient to claim that: "These observations strongly suggest that ATP is produced by the lysosome bound GAPDH/PGK and more effectively supports vesicular proton uptake than free cytosolic ATP." That these enzymes are active is neither surprising nor informative (6E).

According to the review's comment, we observed that GAPDH and PGK are associated with lysosomal vesicle membranes peripherally by ionic bonds. At 0.8 M NaCl, GAPDH and PGK were found to remain associated with lysosomal vesicle membranes, suggesting that GAPDH and PGK are strongly associated with lysosomal vesicle membranes.

GAPDH is mainly cytosolic, it has also been reported to be localized in specific subcellular compartments including the cytoskeleton, membranes and fusion vesicles by palmitoylation. PGK is also modified by palmitoylation, suggesting that these glycolytic enzymes attached to lysosome by modification. In this paper, we described that ATP is produced by the lysosome-bound GAPDH/PGK complex, which should more effectively supports vesicular proton uptake than free cytosolic ATP.

6.. This reviewer recognizes the substantial amount of additional work involved in producing and analyzing the double conditional KO mouse (ATG5/p32). However, the results of the dcKO add little of substance to the study.

According to review's comments, we deleted the result from ATG5/p32 double knockout mice in this resubmitted paper. Now we investigated the mechanism of double knockout heart and prepared another paper in near future.

7.. In summary, the study adds to a growing body of work showing that mitochondrial dysfunction impacts other areas of cell biology and metabolism. A p32 KO with impaired mitochondrial translation/OXPHOS displays lysosome abnormalities

stemming from disturbed NAD metabolism. Concordantly, lysosomal acidification is corrected by NMN supplementation in p32 KO MEFs. The study does seem to lose its way, towards the end and culminates in a major experiment that brought little reward. The obvious question of whether NMN or related compound is of benefit to the p32 KO heart is not addressed.

According to review's comments, we added the comment in Discussion, "NAD⁺ levels increased in animal hearts by NMN addition led to increased Sirt1 activity and protection against ischemia/reperfusion injury (Yamamoto et al, 2014). Thus, NMN treatment is an effective way to activate autophagy in cardiomyocytes and improve heart health and longevity in mouse models. In future studies, we will investigate whether NMN treatment improves the lifespan of mitochondrial translation-deficient heart." Now we performed whether NMN treatment improves p32cKO heart function, however, it takes a long time to achieve results. In this resubmitted paper, we focus on the mechanism how mitochondria translation deficiency decreased lysosomal function.

Other comments and questions:

8.. As the authors note Yamada et al. found that p62 accumulated around mitochondria in Drp1KO hepatocytes and that it was co-localized with ubiquitin and LC3 (Yamada et al., 2018). Taken together with the current findings the implication is that various forms of mitochondrial dysfunction cause p62 accumulation and impede autophagy. This means the initial part of the study of the p32 KO did not yield novel findings.

We agreed the review's comment. We deleted that sentence regarding to p62 in Yamada's work in this resubmitted paper.

9.. The starting hypothesis is in one case weak. Mitochondria with "internal collapse" (i.e. - cristae loss) is a well understood feature of mitochondrial respiratory chain dysfunction. The mitochondria are larger because the lack of cristae junctions decreases the surface area of inner membrane that can 'fit into' a mitochondrion of a given volume). Hence, this is no reason to speculate that mitophagy is impaired or repressed.

We agreed the review's comment that mitochondrial dysfunction and the morphological collaps in p32 cKO heart are not main cause for autophagic impair. in this resubmitted paper, we focus how mitochondrial translation deficiency induces loss of lysosomal

function, leading to autolysosomal defect in this resubmitted paper.

10.. TEM of Fig. 1A shows 1 abnormal mitochondrion in the KO. Many more sections need to be disclosed.

According to review's comment, we added the more Figure in Fig EV1A in this resubmitted paper.

11.. "The size of the p62 and LC3 ring staining pattern was similar to that of the giant mitochondria determined by electron microscopy analysis (Figure 1A), suggesting that LC3-positive autophagosomes sequester damaged mitochondria in p32cKO heart." More examples of each juxtaposed are needed to justify this conclusion.

According to review's comment, we added the data of p62 and LC3 staining and TEM in Expanded View EV1A and EV1B-1C in this resubmitted paper. We demonstrated that "LC3-positive autophagosomes sequester damaged mitochondria in the p32cKO heart". (In Result section)

12.. "fractionation of mitochondria and lysosomes from the membrane fraction demonstrated the Nmnat3 expression in both lysosomal and mitochondrial fractions (Figure 3D)." Expression is not the correct word here, re-write.

According to review's comments, we delete the localization of Nmnat3 in resubmitted paper. We found that Nmnat3 is localized in cytoplasm (cytosol and membrane fraction) in mouse heart (Fig EV2A and Fig 3C).

13.. The nmnat2 data in fig 4B,C should move to supplemental data, as it is better to lead with nmnat3 (as it is more relevant to the heart and the data are better).

According to review's comments, we moved the nmnat2 data in Expanding view (EV3) data in this resubmitted paper and we will focus the nmnat3 in this resubmitted paper.

14.

Introduction:

The introduction is fragmented, the subject switches from p32 to NAD to lysosomes without any links.

According to review's comments, we rewrite the Introduction section in this resubmitted paper. (Page 4)

15.. The end of the Introduction needs re-writing the final paragraph describes ATG5 deficient mice that are a minor part of this study and ends with a sweeping (ungrammatical) statement that lacks context and alludes to but does not describe the important findings. "In the present study, we explain why mitochondrial dysfunction induced loss of lysosomal function, leading to the autolysosome defect, and provide new insight into the essential nature of NAD⁺ in normal lysosomal function."

According to review's comments, we deleted the sentence about double knockout mice and also delete results from ATG5/p32 double knockout mice in this resubmitted paper.

16.. "It is unclear why cardiomyocyte-specific p32-deficient mice survived for a long time". - this statement is unnecessary and uninformative, better to report the facts and the clear conclusions. The results indicate that the mice can survive with cardiomyopathy for 10 months.

According to review's comments, we deleted the sentence and deleted the results of double knockout mice in this resubmitted manuscript.

17.. Fig S2 translation (typo)

According to review's comments, we correct the typing everywhere

Thank you for submitting your revised manuscript. The study has been seen by two of the original referees, whose comments are shown below

As you can see, referee #1 finds that you have sufficiently addressed his/her questions. However, reviewer #2 still feels that the role of autophagy in p32 knockout cardiomyocytes would need to be further characterized. We have discussed this issue with referee #2, who indicated that s/he may support publication of a revised manuscript in which the role of p32 in autophagy in cardiomyocytes is toned down and the existing literature on p32 in autophagy/mitophagy properly discussed (e.g. PMID: 25909887).

We agree with the referee that these further revisions are essential to pursue publication of this study in The EMBO Journal. Given the overall interest of your study, I would like to invite you to submit a new version of the manuscript revised according to the referee's requests.

Referee #1:

The manuscript by Yagi and colleagues studies the role of localized ATP production in the lysosomal membranes by two glycolytic enzymes. This mechanism is impaired in cells and tissues lacking a protein necessary for translation of mtDNA-encoded genes, which results in primary mitochondrial deficiency and secondary lysosomal perturbations.

The effect of the mitochondrial malfunction on the lysosomes is akin to the observed in other studies. However, the authors show a novel mechanism by which mitochondrial malfunction can trigger lysosomal defects. This happens by activation of HIF1a, which represses Nmnat3, an enzyme involved in the synthesis of NAD⁺. This results in decreased NAD⁺ levels in whole cells, which result in decreased ability of carrying out two steps of glycolysis in the proximity of the lysosomes (and, one assumes, all over the cytoplasm), which presumably results in decreased localized ATP production, thus decreasing the ability of the vATPase to function.

The authors improved markedly the manuscript compared to its earlier submission. My technical concerns were addressed, and I believe the data to be now convincing.

I would recommend that the authors include a few sentences in the discussion about the

importance of localized ATP gradients. They mention it briefly in the context of moving vesicles in axons, but it is an important concept for the model evidenced, and therefore it would be nice to have a little more on the topic.

The figure legends should be checked to make sure they always say the type of material used (brains, MEFs) and the age of the animals, sometimes this is missing.

The method for the salting out of the lysosomal proteins should be detailed.

Otherwise, I congratulate the authors on rounding up an innovative model and turning it into a convincing study.

Referee #2:

In the revised version of the manuscript the authors addressed the majority of the issues raised by the referees. Several points have been nicely and properly addressed in terms of new experimental data and manuscript editing. However, still there are some concerns. Some criticisms have been addressed just deleting sentences and statements from the text. This is acceptable for a limited number of points and mainly minor concerns. In this case there are several examples, which raise even more concerns on the whole manuscript. The authors had a particular attention to the second part of the manuscript regarding the metabolic aspects while the first part, more related to the autophagy process, have been poorly addressed. In the abstract the authors mentioned the "autophagic abnormalities" as the first evidence in the p32-knockout mice. In the discussion, autophagy is mostly ignored. Moreover, the first part of the results is dedicated to study autophagy in p32 KO cardiomyocyte. There are several tools and technique that the author could have employed to properly analyse autophagy flux, mitophagy, lysosome function and dissect at which level the autophagy flux is impaired. For example, a more accurate EM could clarify if the mitochondria surrounded by p62 and Ub are inside an autophagosome.

The manuscript improved; however, there are still several weaknesses that should be addressed.

Referee #1:

Reviewers' Comments to Author

Manuscript #: EMBOJ-2020-105268R-Q

We sincerely appreciate that you provided several critical comments. We are sure that your suggestions were very important for improvement of our manuscript. We added several experiments and extensively improve this revised manuscript according to your comments.

1. I would recommend that the authors include a few sentences in the discussion about the importance of localized ATP gradients. They mention it briefly in the context of moving vesicles in axons, but it is an important concept for the model evidenced, and therefore it would be nice to have a little more on the topic

According to reviewer's comment, we added the sentence in Discussion in this revised paper. (Page 16)

“GAPDH is localized to vesicles via a huntingtin-dependent mechanism and is transported by fast-moving vesicles in axons. These specially localized glycolytic mechanism can provide constant energy for the continuous movement of vesicles over long distances within axons.”

- 2.The figure legends should be checked to make sure they always say the type of material used (brains, MEFs) and the age of the animals, sometimes this is missing.

According to reviewer's comment, we changed the Figure legends in this revised manuscript.

We checked and added the age of the WT and p32cKO heart and showed the cell type such as MEF,. p32KO MEF cells or mouse 3T3-L1 cells.

- 3.. The method for the salting out of the lysosomal proteins should be detailed.

According the comments, we changed the sentence in Material and Method in Page 21

“GAPDH and PGK binding to lysosomes

For analysis of GAPDH and PGK binding to lysosomal membranes, **purified** lysosomal fractions (**Fraction No.2**) were divided into 5 parts and resuspended in homogenization buffer (250 mM sucrose, 20 mM HEPES-KOH, pH 7.4, 1 mM EDTA) containing **each**

0, 0.1, 0.2, 0.4, 0.8 M NaCl. The mixtures were incubated for 30 min on ice and subjected to ultracentrifugation at 68,000 rpm for 1 h at 4°C in a fixed angle rotor (TLA120.2; Beckman Coulter). The pellet (lysosomal membrane) were resuspended in 50 µL of SDS-PAGE sample buffer and resolved by SDS-PAGE. The supernatant (protein off lysosomal membrane) were added 20% Trichloroacetic acid (TCA) to extract protein for 30 minutes and centrifuged at 15,000 rpm for 10min at 4°C. The pellet (extract protein) were washed with cold acetone, centrifuged at 15,000rpm for 5min at 4°C. The TCA/acetone precipitate was also suspended in 50 µL of SDS-PAGE sample buffer and resolved by SDS-PAGE. ”

Referee #2:

Manuscript #: EMBOJ-2020-105268R-Q

In the revised version of the manuscript the authors addressed the majority of the issues raised by the referees. Several points have been nicely and properly addressed in terms of new experimental data and manuscript editing. However, still there are some concerns. Some criticisms have been addressed just deleting sentences and statements from the text. This is acceptable for a limited number of points and mainly minor concerns. In this case there are several examples, which raise even more concerns on the whole manuscript. The authors had a particular attention to the second part of the manuscript regarding the metabolic aspects while the first part, more related to the autophagy process, have been poorly addressed. In the abstract the authors mentioned the "autophagic abnormalities" as the first evidence in the p32-knockout mice. In the discussion, autophagy is mostly ignored. Moreover, the first part of the results is dedicated to study autophagy in p32 KO cardiomyocyte. There are several tools and technique that the author could have employed to properly analyse autophagy flux, mitophagy, lysosome function and dissect at which level the autophagy flux is impaired. For example, a more accurate EM could clarify if the mitochondria surrounded by p62 and Ub are inside an autophagosome.

The manuscript improved; however, there are still several weaknesses that should be addressed.

Editor's Comment

However, reviewer #2 still feels that the role of autophagy in p32 knockout cardiomyocytes would need to be further characterized. We have discussed this issue with referee #2, who indicated that s/he may support publication of a revised manuscript in which the role of p32 in autophagy in cardiomyocytes is toned down and the existing literature on p32 in autophagy/mitophagy properly discussed (e.g. PMID: 25909887).

We sincerely appreciate that you provided several critical comments. We are sure that your suggestions were very important for improvement of our manuscript.

According to review's and editor's comments, we delete some sentence and added the New sentence in this revised manuscript.

In Introduction (Page 4)

“It was reported that p32 under starvation conditions by regulating ULK1 stability, and showed a crucial role of the p32-ULK1-autophagy axis in coordinating stress response, cell survival and mitochondrial homeostasis (Jiao *et al*, 2015).”

In Result section (Page 5)

“It was reported that p32 is involved in autophagy by ULK axis (Jiao *et al.*, 2015), we first investigated the expression of autophagic molecule. “

In Discussion (Page 14)

“It was reported that the chaperone-like protein p32 is an important regulator of ULK1 stability by forming a complex with ULK1 (Jiao *et al.*, 2015). P32 knockdown in HeLa cells significantly impaired the clearance of starvation-induced autophagy flux and damaged mitochondria caused by mitochondrial uncoupler. We also found that p32 knockout in heart showed increased phosphorylation of ULK1 Ser757 and p62 Ser403 and also found that autophagic abnormalities, such as p62 accumulation, LC3 around broken mitochondria, however, we did not identify which level the autophagy flux is impaired in p32 knockout heart. In future studies, we will investigate which stage of autophagy flux is involved in p32-deficient heart. “

2nd Revision - Editorial Decision**10th Nov 2020**

Thank you for submitting your revised manuscript. Please accept my sincerest apologies for the delay in getting back to you. I have now checked all files and found that there are a few editorial issues concerning the text and the figures that I need you to address before we can officially accept your manuscript.

3rd Authors' Response to Reviewers**21st Nov 2020**

All editorial issues have been addressed

3rd Revision - Editorial Decision**8th Dec 2020**

I am pleased to inform you that your manuscript has been accepted for publication in The EMBO Journal.

Corresponding Author Name: Takeshi Uchiumi

Manuscript Number: EMBOJ-2020-105268